# What is the true discharge rate and pattern of the striatal projection neurons in Parkinson's disease and Dystonia?

Dan Valsky[1,2], Shai Heiman Grosberg[1], Zvi Israel[3], Thomas Boraud[4,5,6], Hagai Bergman[1,2,3], Marc Deffains[4,5]*

[1]Department of Medical Neurobiology, Institute of Medical Research Israel - Canada (IMRIC), The Hebrew University - Hadassah Medical School, Jerusalem, Israel; [2]The Edmond and Lily Safra Center for Brain Sciences, The Hebrew University, Jerusalem, Israel; [3]Department of Neurosurgery, Hadassah University Hospital, Jerusalem, Israel; [4]University of Bordeaux, UMR 5293, IMN, Bordeaux, France; [5]CNRS, UMR 5293, IMN, Bordeaux, France; [6]CHU de Bordeaux, IMN Clinique, Bordeaux, France

**Abstract** Dopamine and striatal dysfunctions play a key role in the pathophysiology of Parkinson's disease (PD) and Dystonia, but our understanding of the changes in the discharge rate and pattern of striatal projection neurons (SPNs) remains limited. Here, we recorded and examined multi-unit signals from the striatum of PD and dystonic patients undergoing deep brain stimulation surgeries. Contrary to earlier human findings, we found no drastic changes in the spontaneous discharge of the well-isolated and stationary SPNs of the PD patients compared to the dystonic patients or to the normal levels of striatal activity reported in healthy animals. Moreover, cluster analysis using SPN discharge properties did not characterize two well-separated SPN subpopulations, indicating no SPN subpopulation-specific (D1 or D2 SPNs) discharge alterations in the pathological state. Our results imply that small to moderate changes in spontaneous SPN discharge related to PD and Dystonia are likely amplified by basal ganglia downstream structures.

*For correspondence:
marc.deffains@u-bordeaux.fr

Competing interests: The authors declare that no competing interests exist.

## Introduction

Parkinson's disease (PD) and Dystonia are two of the most common movement disorders, and have a wide spectrum of etiologies and clinical presentations. To date, the pathophysiology of PD and Dystonia is still debated. Traditionally, PD is attributed to the degeneration of the midbrain dopaminergic neurons which also disrupts other neuromodulary systems, including the cholinergic, serotoninergic and histaminergic systems (*Bolam and Ellender, 2016*; *Deffains and Bergman, 2015*; *Fox et al., 2009*). Although there is no degeneration of the midbrain dopaminergic neurons in Dystonia, an imbalance between the midbrain dopaminergic and striatal cholinergic systems (*Aosaki et al., 2010*; *Benarroch, 2012*; *Bonsi et al., 2011*; *Pisani et al., 2007*), as well as cerebellum dysfunction (*Fremont et al., 2017*; *Helmich, 2018*; *Helmich et al., 2012*; *Tewari et al., 2017*; *Wu and Hallett, 2013*) are present in both PD and Dystonia. Nevertheless, basal ganglia (BG) dysfunction is still regarded as the source of the cardinal symptoms of both diseases (*Mink, 1996*; *Wichmann, 2018*) and deep brain stimulation (DBS) in the BG (subthalamic nucleus, STN and internal segment of the globus pallidus, GPi) is an effective invasive treatment for both diseases (*Limousin et al., 1998*; *Moro et al., 2017*; *Odekerken et al., 2016*; *Ostrem et al., 2017*).

The striatum (i.e., the main input structure of the BG network) is the main recipient of midbrain dopaminergic neurons (*Menegas et al., 2015*) and also receives di-synaptic cerebellar projections

(*Bostan et al., 2013*; *Hoshi et al., 2005*). Most [~95% and ~70–80% in rodents and primates, respectively (*Graveland et al., 1985*; *Petryszyn et al., 2018*; *Petryszyn et al., 2014*)] striatal neurons are medium spiny projection neurons (SPNs) that receive afferents from the cortex and the thalamus, and together with the STN neurons, innervate the central (i.e., external segment of the globus pallidus, GPe) and output (i.e., GPi and substantia nigra reticulata, SNr) BG structures (*Albin et al., 1989*; *Gerfen et al., 1990*). Therefore, alterations in striatal signaling disrupt normal BG activity and may lead to the manifestation of the motor and non-motor symptoms of PD and Dystonia (*Albin et al., 1989*; *Bergman et al., 1990*; *Gerfen et al., 1990*).

Dysregulation of BG activity may consist of changes in discharge rate (*Albin et al., 1989*; *Gerfen et al., 1990*). Unlike in all other BG structures, extracellular recordings of spiking activity in non-human primates (NHPs) reveal that SPNs have a very low discharge rate (~1–2 Hz at rest) and are phasically active (i.e. emit short bursts) around relevant behavioral events (*Crutcher and DeLong, 1984*; *Deffains et al., 2010*; *Kimura et al., 1990*). However, previous studies of the BG in the NHP model of PD have mainly focused on the STN, GPe and GPi which are structures with a high frequency tonic discharge (i.e. 25–70 Hz at rest) (*Deffains et al., 2016*; *Filion and Tremblay, 1991*; *Wichmann and DeLong, 2003*). These studies have reported excessive GPi/SNr discharge rates in PD that lead to an increase in BG inhibitory outputs to the thalamus and the frontal cortex motor areas (*Wichmann and DeLong, 2003*). Conversely, it is assumed that Dystonia (and other hyper-kinetic states) are characterized by reduced GPi/SNr activity (*Guehl et al., 2009*).

More recent studies of the BG in animal (rodent and NHP) models of PD and human patients (undergoing DBS procedures) have dealt with changes in discharge patterns and synchronization. Parkinsonism-related β oscillations have been observed in local field potentials (LFPs) recorded in all BG structures, including the striatum (*Deffains et al., 2016*; *Kondabolu et al., 2016*; *Lemaire et al., 2012*; *Singh and Papa, 2019*). Similarly, low frequency (4–12 Hz) LFP oscillations have been recorded in the BG network of dystonic patients (*Piña-Fuentes et al., 2018*; *Silberstein et al., 2003*). Finally, synchronous β oscillations are commonly observed in the spiking activity of the STN, GPe and GPi of MPTP-treated monkeys (*Deffains et al., 2018*; *Deffains et al., 2016*; *Soares et al., 2004*) and PD patients (*Brown, 2003*; *Kuhn et al., 2005*; *Moshel et al., 2013*; *Zaidel et al., 2010*).

Nevertheless, direct evidence of abnormal activity of the striatal SPNs is still elusive. Previous studies utilizing the NHP model of PD reported striking increases (~15 fold increase from the normal discharge rate of ~1–2 Hz) in the firing rate of the SPNs subsequent to striatal dopamine depletion and the induction of parkinsonism (*Liang et al., 2008*; *Singh et al., 2016*). The same research group also reported a high discharge rate of SPNs recorded in PD and dystonic patients (~30 Hz and 9 Hz, respectively) (*Singh et al., 2016*). They also found a significant change in the firing pattern of striatal neurons, with many SPNs exhibiting bursting activity in PD patients and MPTP monkeys as compared to a smaller fraction in patients with Dystonia (*Singh et al., 2016*). Finally, the SPNs of patients with an essential tremor (ET, regarded as a non-BG disorder) have a very low discharge rate (~2 Hz) and no tendency to burst, as reported in normal NHPs (*Liang et al., 2008*; *Singh et al., 2016*; *Singh et al., 2015*).

These spectacular changes in the discharge rate and pattern of SPNs in the NHP model of PD run counter results obtained in our research group. We recorded the activity of SPNs and other BG neurons in Vervet monkeys before and after systemic 1-methyl-4-phenyl-1,2,3,6-tetrahydropyridine (MPTP) treatment and the induction of severe parkinsonian symptoms. Although we found robust changes in the discharge properties (rate, pattern and synchronization) of the STN, GPe and BG output structures, we did not observe any difference in the discharge rate (~2–3 Hz) or pattern of SPNs in the MPTP-treated monkeys in comparison to the recordings in the same monkeys before MPTP treatment (*Deffains et al., 2018*; *Deffains et al., 2016*; *Deffains and Bergman, 2019*).

Extracellular recordings of SPN spiking activity of anesthetized (*Chen et al., 2006*; *Mallet et al., 2006*; *Sharott et al., 2017*) and awake (*Chen et al., 2006*; *Kish et al., 1999*) rats before and after striatal dopamine depletion by 6-hydroxydopamine (6-OHDA) treatment have revealed a significant, but very slight increase in the SPN discharge rate. Rodent studies make it possible to differentiate between SPNs expressing D1 and D2 dopamine receptors. A significant imbalance (but see *Ketzef et al., 2017*) in the discharge rate and calcium dynamics of D1 and D2 SPNs was observed in the dopamine-depleted striatum (*Mallet et al., 2006*; *Parker et al., 2018*; *Sharott et al., 2017*). In particular, D2 SPNs increased their discharge (*Mallet et al., 2006*; *Sharott et al., 2017*) and were

also prone to being entrained to parkinsonian β oscillations (*Sharott et al., 2017*). Nevertheless, the absolute increase in the discharge rate even of the D2 SPNs was still modest (from ~0.5 to~2.8 Hz) (*Sharott et al., 2017*). Moreover, recent studies have reported no significant increase in the low discharge rate of either SPN subpopulation in striatal dopamine-depleted mice (*Ketzef et al., 2017*; *Maltese et al., 2019*).

Since only one research group has recorded drastic increases in the spontaneous spiking activity in the striatum of PD patients in comparison to either dystonic patients or the normal levels of striatal activity reported in healthy animals (*Singh et al., 2016*), we examined our human patient data to determine whether we could report similar changes. Our study goes beyond previous reports (*Singh et al., 2016*) by utilizing a machine learning algorithm to automatically detect striato-pallidal border, and applying objective methods for the identification of units and the quantification of their isolation quality and stationarity. Although extracellular recording methods cannot discriminate between the spiking activity of striatal D1 and D2 SPNs, we assumed that if only one population of striatal neurons was strongly affected by the pathological state, we should observe a significantly higher discharge rate of SPNs than that reported in normal NHPs and ET patients, and/or distinct clusters of SPN activity in our patients. Using this data-driven automated approach, we carefully examined spontaneous single-unit activity from the posterior putamen (i.e., the sensorimotor domain of the striatum) of both PD and dystonic patients undergoing GPi-DBS surgeries and compared their discharge rates and patterns.

## Results

### Database and spike sorting results

In this study, 93 microelectrode trajectories (48 in PD and 45 in Dystonia) were used, yielding a total of 933 and 718 microelectrode recording (MER) segments within the posterior putamen (i.e., posterior to the anterior commissure, *Figure 1* and *Video 1*) of patients suffering from PD and Dystonia, respectively. Microelectrode trajectories from patients suffering from non-genetic (N = 27) and genetic (N = 18) Dystonia were pooled, since the results did not differ statistically. Applying automatic spike sorting on these MER segments identified 5237 units (2917 in PD and 2320 in Dystonia). The isolation score (*Joshua et al., 2007*) - ranging from 0 (i.e., highly noisy) to 1 (i.e., perfect isolation) - was calculated for each of these units. Moreover, since injury potentials are often observed during extracellular microelectrode recordings of striatal activity (*DeLong, 1971*), the stationarity of the firing rate and spike amplitude of the well-isolated units (isolation score $\geq 0.6$) was assessed.

All surgeries were carried out while the patients were fully awake (no sedation or anesthesia) and the PD patients were off dopaminergic medication (overnight washout >12 hr). The DBS target was the ventro-posterior-lateral portion of the GPi for all patients, and trajectory angles were only slightly modified according to patient's anatomy. To graphically illustrate the recording locations in the striatum, we employed visualization of group-based microelectrode track trajectories in both PD and Dystonia (*Figure 1*, see also *Video 1*). All recordings were regularly sampled in space within the striatum (see Materials and methods). All patients provided their written informed consent and the study was approved by the Institutional Review Board of Hadassah Hospital in accordance with the Helsinki Declaration (reference code: 0168–10-HMO).

*Figure 2A* depicts examples of the striatal spiking activity recordings (left and middle). The right plots display the superimposed waveforms of the extracellular action potentials of the sorted units. The units are ordered as a function of their isolation score (*Joshua et al., 2007*). The median and mean discharge rates of all the recorded units with an isolation score greater than or equal to one of the 10 evenly spaced values between 0 and 0.9 were calculated (*Figure 2B*). A significant negative correlation between neuronal discharge rate and isolation score was found in the striatum of the PD (Pearson's r = $-0.87$ and $-0.89$ for median and mean discharge rate, respectively, p<0.01, *Figure 2B*) and dystonic (Pearson's r = $-0.89$ and $-0.85$ for median and mean discharge rate, respectively, p<0.01, *Figure 2B*) patients, indicating that the units with the lowest isolation score had the highest discharge rates.

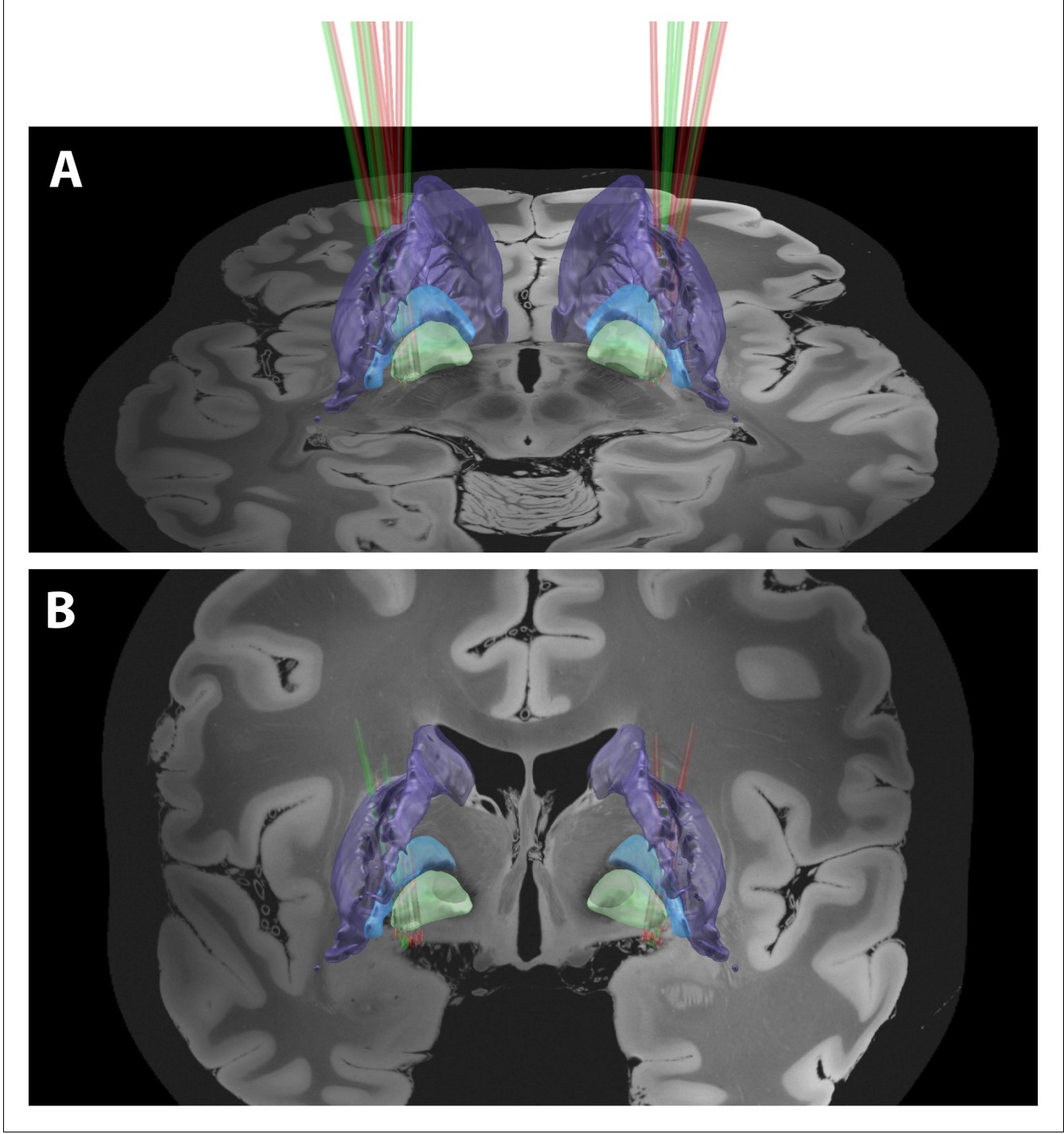

**Figure 1.** A 3D rendering of group-based microelectrode track trajectories in the posterior putamen. (**A**) Posterior view of the microelectrode track trajectories from both PD and dystonic patients. An axial image from the normalized scan, at the level of the rostral midbrain, is shown as a backdrop. The image was acquired on a 7T MRI scanner with a 100 μm T1 scan of an ex-vivo human brain. The definition of the striatum (purple), GPe (light blue) and GPi (light green) boundaries are defined by the DISTAL atlas. (**B**) Same as (**A**) but with a coronal image. A total of 36 microelectrode track trajectories, 13 in PD (green) and 23 in Dystonia (red), are visualized for four PD and six dystonic patients.

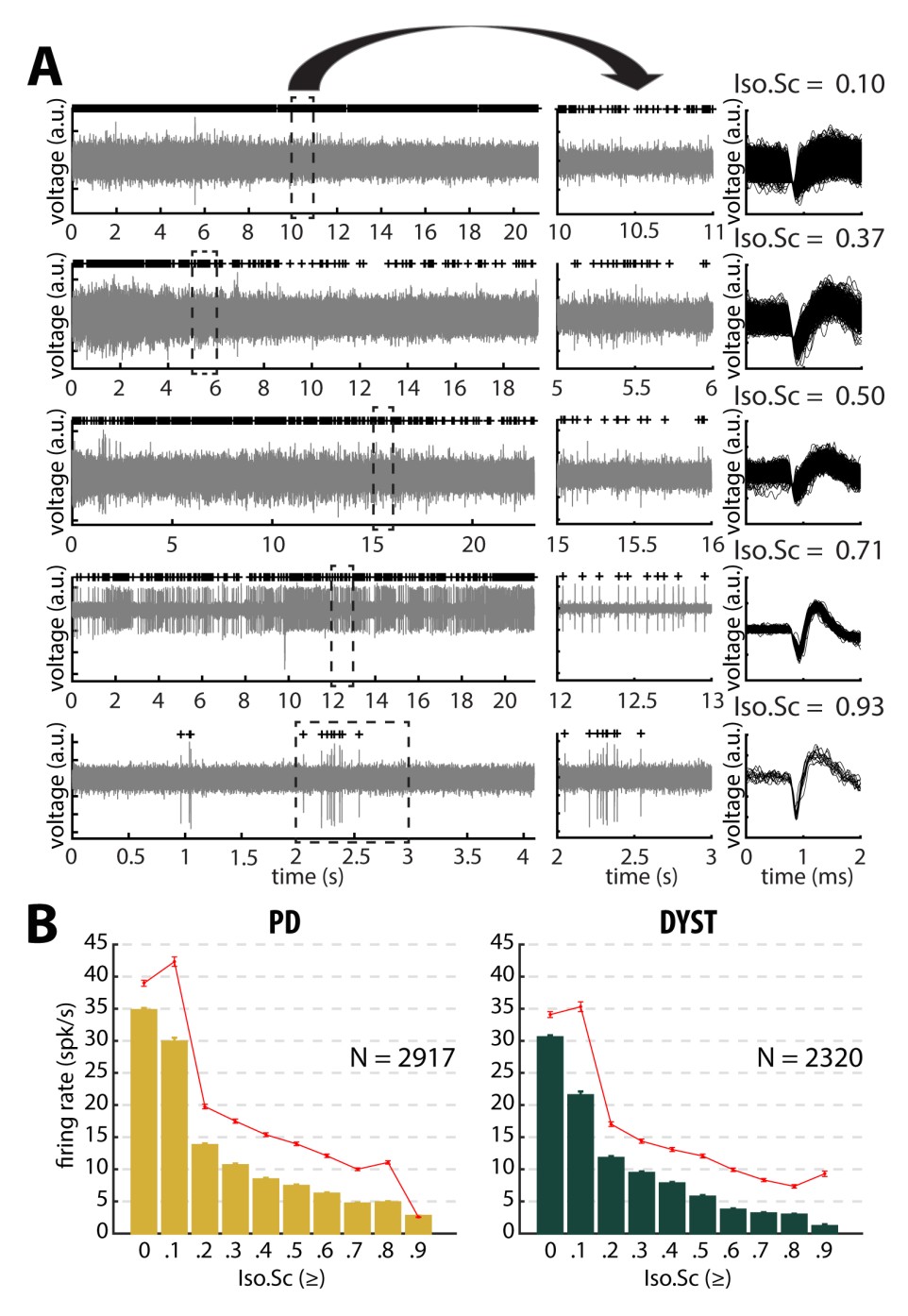

**Figure 2.** Striatal firing rate in parkinsonian (PD) and dystonic (DYST) patients decreases as isolation quality of the units increases. (**A**) Examples of full-length (left panels) and 1 s (middle panels) striatal spiking recordings. Above the spiking activity is the digital display of the detection (spike train) of the sorted unit from the recording. Spike waveforms are superimposed (right panels). Isolation score (Iso.Sc) indicates the isolation quality of the sorted unit. (**B**) Evolution of the firing rate as a function of the isolation score. Each bar indicates the median firing rate of the sorted units with an isolation score greater than or equal to a certain value (i.e., the bin labelled 0.8 contains all the units with an isolation score ≥0.8). Error bars represent MADs (i.e., median absolute deviations). Means ± SEMs (i.e., standard errors of the mean) are shown in red. N is the number of sorted units.

The online version of this article includes the following source data for figure 2:

**Source data 1.** Firing rate and isolation score of all units in PD.

**Source data 2.** Firing rate and isolation score of all units in Dystonia.

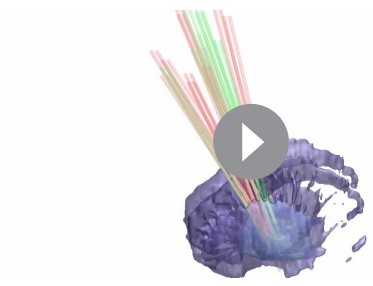

**Video 1.** 3D rotating view of group-based microelectrode trajectories.
https://elifesciences.org/articles/57445#video1

## No drastic increase in the firing rate of the striatal neurons of PD and dystonic patients

The median firing rate of all the sorted striatal units was relatively high (~30–40 Hz) in both diseases (*Figure 3*, first column) and consistent (at least for PD) with the values reported by *Singh et al., 2016*. Similarly, the firing rate of the dystonic patients was significantly lower than the PD firing rate (Mann-Whitney U-test, p<0.001, *Figure 3—figure supplement 1*, first column). Nevertheless, this significant difference disappeared and the striatal firing rate decreased dramatically when only comparing the firing rate of the well-isolated units (isolation score ≥0.6) (*Figure 3* and *Figure 3—figure supplement 1*, first column). Hence, there was no drastic change in the striatal firing rate of the well-isolated units of the PD and dystonic patients.

Inspection of the spike train of the well-isolated units (isolation score ≥0.6) revealed that although the units had been graded as high-isolation score units, their spike train could be non-stationary (*Figure 4*). To assess the stationarity of the spike trains, we examined the temporal (linear) evolution of their firing rates and spike amplitudes (see Materials and methods). *Figure 4A* shows three representative examples of well-isolated non-stationary units (left panels) and three representative examples of well-isolated stationary units (right panels) from the striatum of PD patients. Here, we used the slopes of the linear regression lines (reflecting the temporal evolution of the firing rate and spike amplitude of the well-isolated units) to define stationary units (*Figure 4B and C*). Interestingly, the firing rate slope values and spike amplitude slope values of the well-isolated units were uniformly distributed, thus indicating that these two spiking features were independent in both PD and Dystonia (*Figure 4D and E*). This analysis also showed that the non-stationary units (i.e., data points outside the shaded areas) did not necessarily exhibit both a non-stationary firing rate and a non-stationary spike amplitude (*Figure 4D and E*). For both PD and Dystonia, the firing rate of the stationary units was significantly lower than the firing rate of the non-stationary units (Mann-Whitney U-test, p<0.01 and 0.001 for PD and dystonic patients, respectively, *Figure 4F*). Thus, inclusion of non-stationary units could have erroneously inflated the striatal firing rate in PD and dystonic patients. Removing the non-stationary units reduced the striatal firing rate of the well-isolated units in the striatum of both PD and dystonic patients (*Figure 3* and *Figure 3—figure supplement 1*, first column). Moreover, the striatal firing rate remained similar between both diseases after this maneuver.

Equally important, the mean values of the firing rate (*Figure 3—figure supplement 1*, first column) were systematically higher than the median values (*Figure 3*, first column), suggesting that whatever the quality of the isolation and the stationarity of the striatal units, the distributions of the firing rate were not normal and were skewed to the right. *Figure 5* depicts the distributions of the firing rate of all and only the well-isolated units (*Figure 5A and B*, respectively). For both PD and Dystonia, and for all levels of spike isolation, the distributions of the firing rate of the SPNs were not normally distributed, but rather were strongly skewed to the right (*Figure 5*). We therefore used the median (*Figure 3*) rather than the mean (*Figure 3—figure supplement 1*) to represent the central moment of the distributions of the discharge properties (including the discharge rate) of the SPNs. The median ± MAD (median absolute deviation) of the discharge rate was 3.85 ± 1.18 Hz and 2.55 ± 0.81 Hz for the well-isolated stationary SPNs in PD and Dystonia respectively - i.e., in the same range as reported for the controls in animal studies and with no significant difference between PD and Dystonia (Mann-Whitney U-test, p>0.05, *Figure 4F*).

## No evidence for bursty patterns in the striatal spiking activity of PD and dystonic patients

To characterize the pattern of the spike trains (i.e., irregular, periodic or bursty) and compare them between PD and dystonic patients, we examined the time interval histograms (TIHs) of the inter-

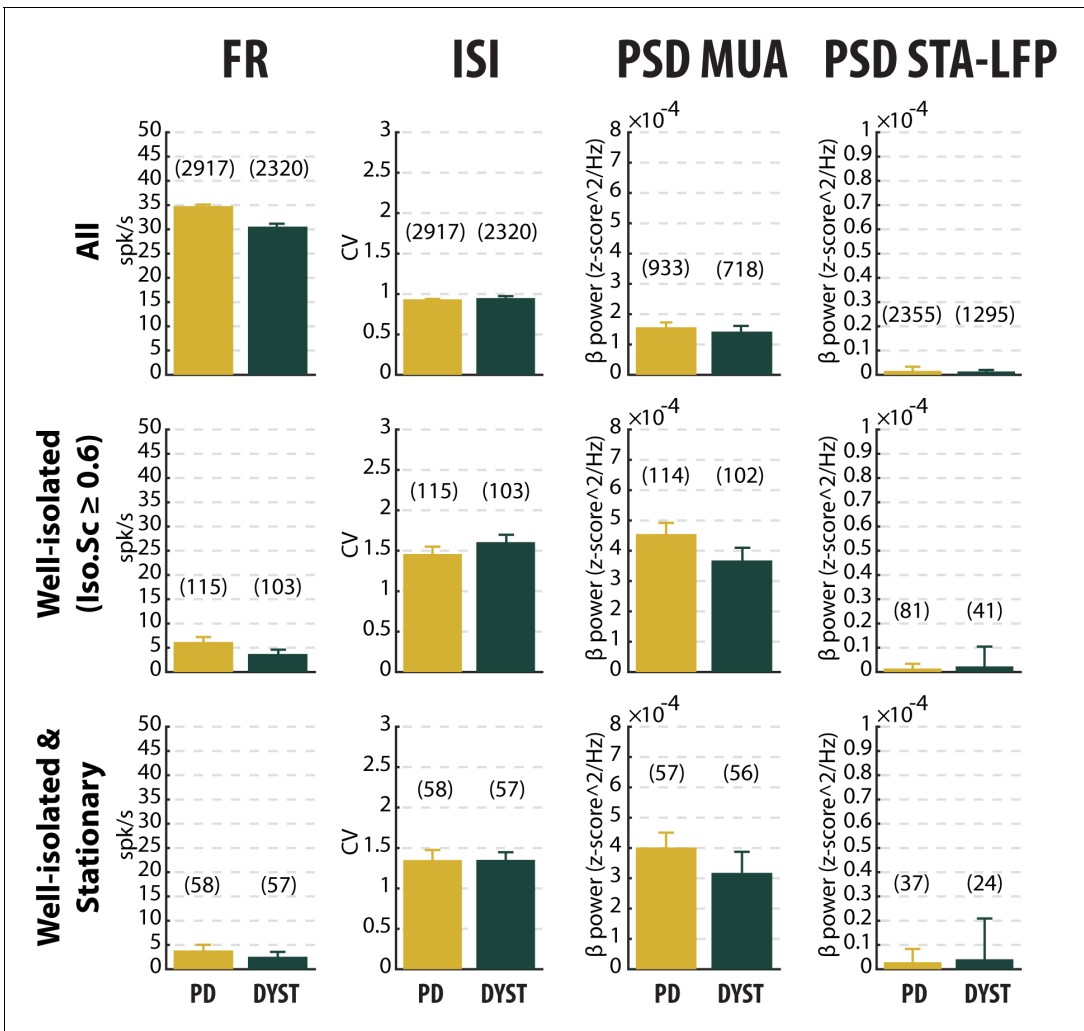

**Figure 3.** No drastic or specific change in the features (median ±MAD values) of striatal neuronal activity in parkinsonian (PD) and dystonic (DYST) patients. Comparison of the firing rate (first column), the coefficient of variation (CV) of the inter-spike interval (ISI) (second column), the β power of the multi-unit activity (MUA, spiking activity) recorded in the vicinity of a sorted unit (third column) and the β power of the spike-triggered average (STA) of the local field potential (LFP) (fourth column) when considering all (upper panels), only the well-isolated (middle panels) and only the well-isolated stationary (lower panels) units. Since each signal was Z-score normalized (using its mean and SD) prior to PSD calculations, the β power of both the MUA (third column) and STA-LFP (fourth column) are expressed in z-score2/Hz. Each bar indicates the median value and error bars represent MADs. Same y-axis scales as in *Figure 3—figure supplement 1* and *Figure 3—figure supplement 5*. Numbers in parentheses indicate the sizes of the samples. Note that the sizes of the samples differ between the columns because (i) the number of sorted units from each MUA ranged from 1 to 5 and (ii) the LFPs were not systematically recorded.

The online version of this article includes the following source data and figure supplement(s) for figure 3:

**Source data 1.** Spiking features of all units in PD.
**Source data 2.** Spiking features of the well-isolated units in PD.
**Source data 3.** Spiking features of the well-isolated and stationary units in PD.
**Source data 4.** Spiking features of all units in Dystonia.
**Source data 5.** Spiking features of the well-isolated units in Dystonia.
**Source data 6.** Spiking features of the well-isolated and stationary units in Dystonia.
**Figure supplement 1.** No drastic or specific change in the features (mean ± SEM values) of striatal neuronal activity in parkinsonian (PD) and dystonic (DYST) patients.
**Figure supplement 2.** Comparison of the spiking features between the putative well-isolated and stationary SPNs and TANs in PD patients.
**Figure supplement 3.** Comparison of the spiking features between the putative well-isolated and stationary SPNs and TANs in dystonic patients.
**Figure supplement 4.** Disease effects on the outliers of the different features of striatal neuronal activity.
**Figure supplement 5.** Median (upper panels) and mean (lower panels) values of the main features of striatal neuronal activity in parkinsonian (PD) and dystonic (DYST) patients.

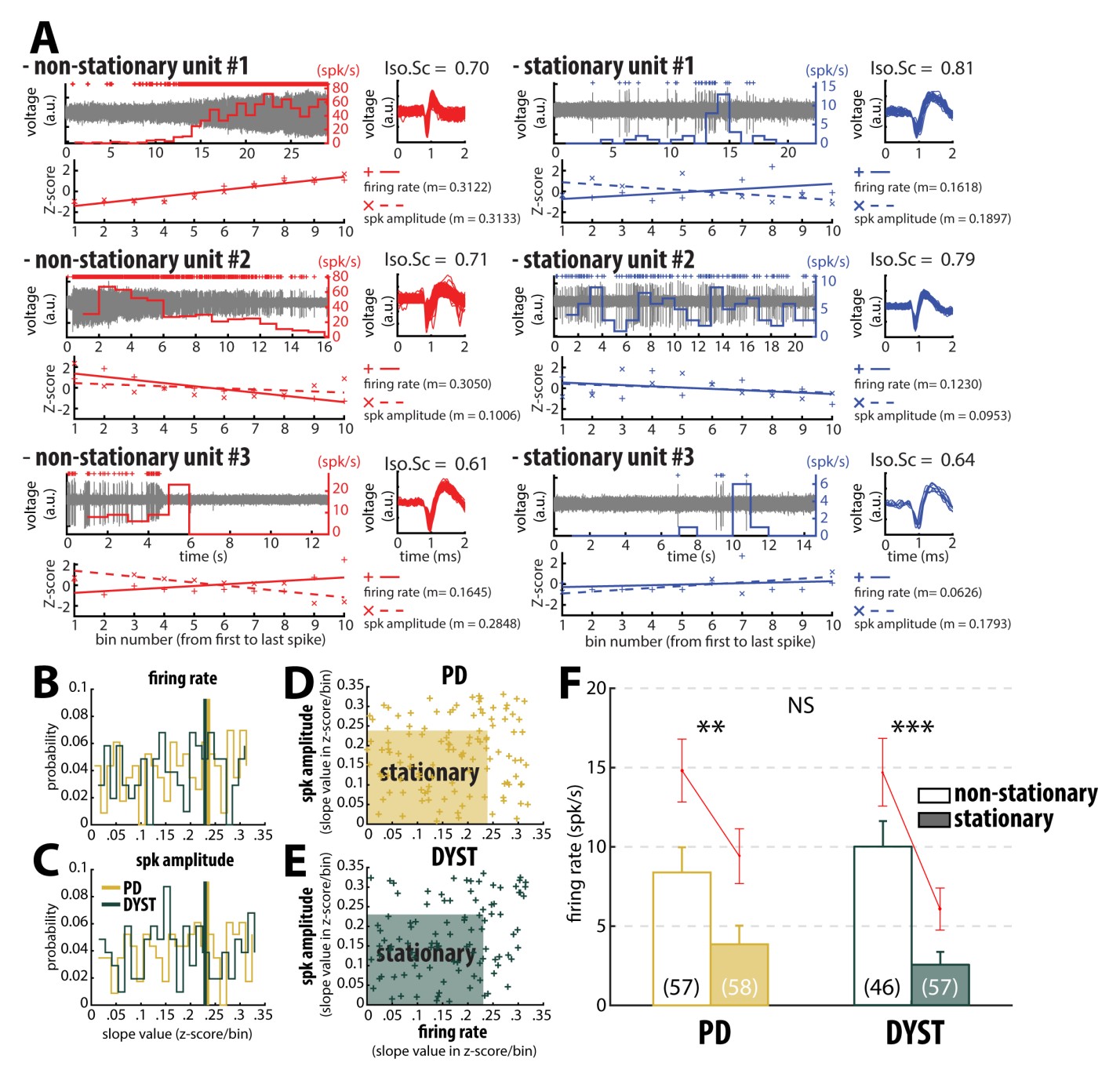

**Figure 4.** Inclusion of non-stationary units can erroneously increase the striatal firing rate in parkinsonian (PD) and dystonic (DYST) patients. (**A**) Examples of well-isolated (non-stationary and stationary) units recorded in the striatum of PD patients. Grey trace is the band-pass filtered signal and depicts the spiking activity. Above the spiking activity is the digital display of the detection (spike train) of the sorted unit. The firing rate of the sorted unit overlays the spiking activity. The spike waveforms extracted from the spiking activity are superimposed and displayed on the right. The isolation score (Iso.Sc) indicates the isolation quality of the identified unit. Below the spiking activity panel is the assessment of the stationarity of the firing rate and spike amplitude. For each spike train, the data comprised between the first and the last spike of the spike train were segmented into 10 equal non-overlapping time bins. Then, the firing rate and the average spike amplitude in each bin were Z-score normalized using the mean and the SD of either the firing rate or the average spike amplitude calculated over the 10 bins. Solid and dotted lines represent the linear regression lines between the firing rate/spike amplitude of the sorted unit and bin number. Slope value (z-score/bin) of the linear regression line (m) was used to assess stationarity. (**B–C**) Distributions of the slope values of the linear regression line for the firing rate and the spike amplitude of the units. Vertical lines indicate the 70th percentile for the two diseases (PD: 0.2363 and 0.2344 z-score/bin for firing rate and spike amplitude, respectively; Dystonia: 0.2292 and 0.2294 z-score/bin for firing rate and spike amplitude, respectively). Units with slope values of the linear regression line for firing rate or spike amplitude greater than

*Figure 4 continued on next page*

*Figure 4 continued*

or equal to the 70th percentile were defined as non-stationary units. (**D–E**) Scatter plots showing the firing rate slope values and spike amplitude slope values of the well-isolated (non-stationary and stationary) units recorded in the striatum of PD and dystonic patients. Shaded areas represent the zones of the stationary units which are delineated by the 70th percentile of both the slope values and spike amplitude slope values. (**F**) Comparison of the firing rate of the non-stationary and stationary units. Each bar indicates the median firing rate of the sorted units. Error bars represent MADs. Means ± SEMs are shown in red. Numbers in parentheses indicate the numbers of non-stationary and stationary units. **, *** and NS indicate significant (p<0.01 and 0.001) and non-significant differences, respectively (Mann-Whitney U-test).

spike intervals (ISIs). Although weak (*Figure 3*, second column), there was a significant difference in the coefficient of variation (CV) of the ISIs of all the sorted striatal units between the PD and dystonic patients (Mann-Whitney U-test, p<0.001, *Figure 3—figure supplement 1*, second column). Nevertheless, as was the case for the striatal firing rate, this difference vanished when only considering the well-isolated or well-isolated stationary units (*Figure 3* and *Figure 3—figure supplement 1*, second column). Again, the mean values of the CV of the ISIs (*Figure 3—figure supplement 1*, second column) were systematically higher than the median values (*Figure 3*, second column), indicating skewed distributions. Moreover, the distributions of TIHs of the ISIs of the well-isolated stationary units matched the Poisson temporal distribution of the spikes (*Figure 6A*). Accordingly, the mean autocorrelograms of the spike trains did not reveal any periodic or bursty firing patterns in the

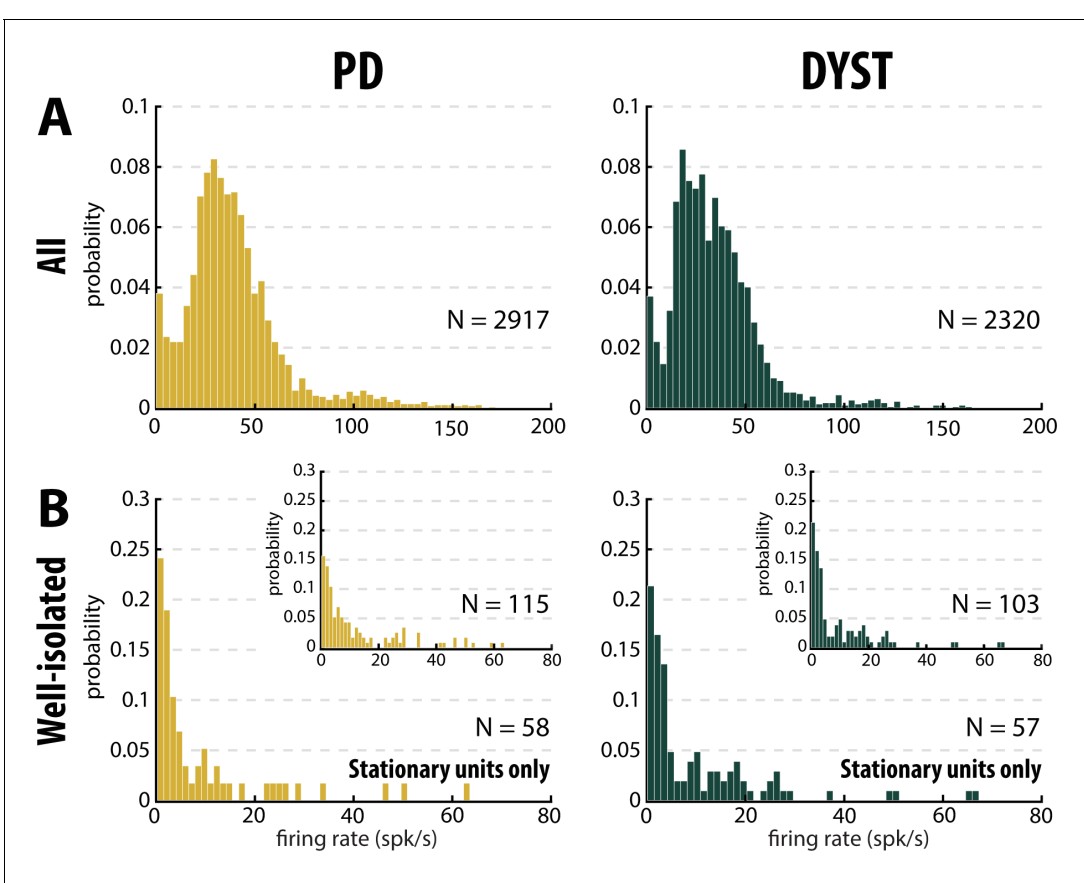

**Figure 5.** Distributions of the striatal firing rate are skewed to the right in parkinsonian (PD) and dystonic (DYST) patients. (**A**) Firing rate of all sorted units regardless of their isolation quality. (**B**) Firing rate of the well-isolated stationary units only. Skewness = 2.34 and 3.94 for PD and dystonic patients, respectively. Insets: Firing rate of the well-isolated units (non-stationary and stationary units pooled). Skewness = 1.73 and 2.36 for PD and dystonic patients, respectively. N is the number of sorted units.

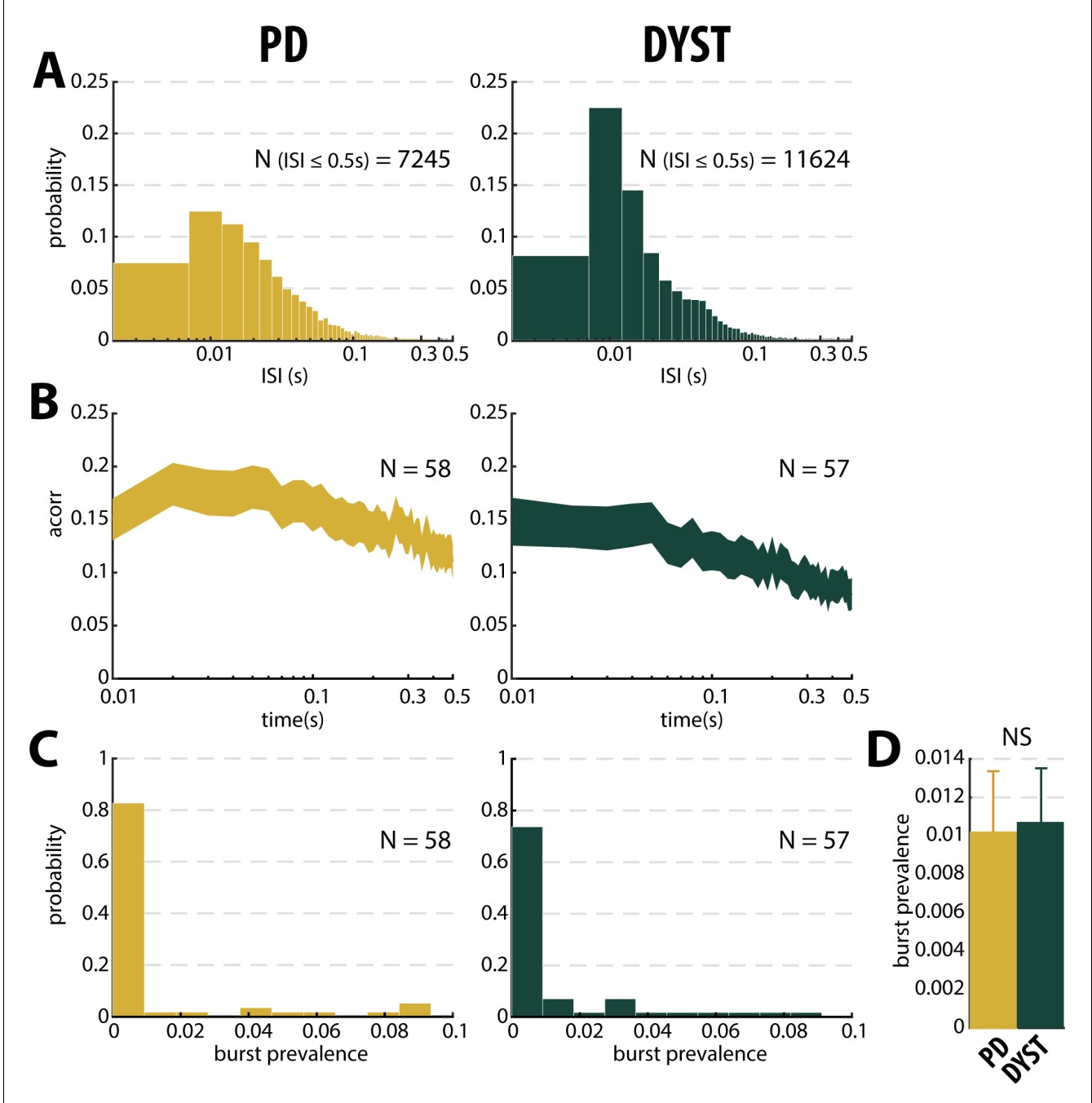

**Figure 6.** No evidence for burst patterns in the striatal spiking activity of parkinsonian (PD) and dystonic (DYST) patients. (**A**) Time interval histograms of the inter-spike intervals (ISI) of the well-isolated (isolation score ≥0.6) stationary units. For better visualization, 191 and 292 ISIs > 0.5 s were removed from the PD and DYST striatal units, respectively. Abscissas are in log scale. (**B**) Average (mean ± SEM) autocorrelograms of the well-isolated stationary units. For each autocorrelogram, values were normalized so that the autocorrelation values ranged from 0 to 1. Abscissas are in log scale. N is the number of well-isolated stationary units averaged. (**C**) Distributions of the values of burst prevalence for the spike train of the well-isolated stationary units. For each unit, episode prevalence represents the probability that the discharge pattern was bursty. (**D**) Mean values of burst prevalence for the well-isolated stationary units recorded in the striatum of parkinsonian and dystonic patients. Error bars represent SEMs. N is the number of well-isolated stationary units averaged. NS: non-significant (Mann-Whitney U-test).

The online version of this article includes the following figure supplement(s) for figure 6:

**Figure supplement 1.** No evidence for burst patterns in the striatal spiking activity of parkinsonian (PD) and dystonic (DYST) patients.

striatal well-isolated stationary units (*Figure 6B*). Similarly, the prevalence of bursts [detected using the Poisson surprise method with the surprise maximization (SM) search algorithm (*Legéndy and Salcman, 1985*), see Materials and methods] was relatively low (*Figure 6C*) and did not differ significantly between PD and dystonic patients (Mann-Whitney U-test, p>0.05, *Figure 6D*).

## Absence of oscillatory spiking activity in the striatum of PD and dystonic patients

Theoretical studies have shown that neural oscillations can emerge at the population level in networks of neurons exhibiting an irregular (i.e., non-oscillatory) discharge pattern and a low firing rate (*Brunel and Hakim, 2008*; *Kopell and LeMasson, 1994*). To overcome the possible confounding effects of the low discharge rate and spatial under-sampling of the striatal SPNs, we investigated multi-unit oscillatory activity rather than single-unit oscillatory activity. Striatal multi-unit activity (MUA) reflects the spiking activity of an ensemble of striatal neurons around a sorted unit.

Comparison of the power spectral densities (PSDs) of the spiking activities recorded in the striatum of the PD and dystonic patients did not reveal any oscillatory phenomena between 3 and 75 Hz, including the β (13–30 Hz) band (*Figure 7A and B*). Accordingly, no significant difference in the β power of the striatal spiking activities (whatever the quality of the isolation and the stationary of the striatal units) was observed between PD and dystonic patients (Mann-Whitney U-test, p>0.05, *Figure 3*, and *Figure 3—figure supplement 1*, third column), Moreover, we found no significant increases of the β (13–30 Hz) power in the spiking activities recorded in the vicinity of the striatal well-isolated stationary units compared to the linearly interpolated baseline β power (Wilcoxon signed rank test, p>0.05, *Figure 7C*), thus supporting the claim of no oscillatory spiking activity in the striatum of PD and dystonic patients.

## No locking between spike and β LFP oscillations in the striatum of PD and dystonic patients

The mean PSD of the mono-polar LFPs recorded in the striatum of PD and dystonic patients (*Figure 8A*) exhibited (ignoring 50 Hz artifacts) two and three distinct peaks, respectively (at 11 and 21 Hz for PD patients and 8, 22 and 30 Hz for dystonic patients, *Figure 8A*, insets). Therefore, we calculated the spike-triggered averages of the LFPs (STAs LFPs). The β power in the LFPs recorded around the time of the spikes of all the striatal sorted units was significantly higher in PD than in dystonic patients (Mann-Whitney U-test, p>0.001, *Figure 3* and *Figure 3—figure supplement 1*, fourth column). However, this difference disappeared when using the spikes of the well-isolated (non-stationary and stationary or only stationary) units (*Figure 3* and *Figure 3—figure supplement 1*, fourth column). Thus, the spiking activity of well-isolated stationary units recorded in the striatum of both PD and dystonic patients failed to lock with 13–30 Hz LFP oscillations (*Figure 8B*).

## TANs are hardly distinguishable from SPNs in our human databases

In order to differentiate between the SPNs and the striatal cholinergic interneurons, the tonically active neurons (TANs, presumably the cholinergic interneurons) were retrospectively identified by three experts (DV, HB and MD) from the well-isolated and stationary striatal units included in both the PD and Dystonia databases. This identification was based on offline visual inspection of their electrophysiological features. The 3D-classifications of the putative SPNs and TANs, using their firing rate, the CV of their ISI and the score of the first principal component (PC1) of their spike waveform did not clearly discriminate them in either PD or dystonic patients (*Figure 3—figure supplement 2A* and *Figure 3—figure supplement 3A*, respectively). The same 3D-classification, but using the score of PC2 or PC3 (instead of PC1) yielded the same qualitative results. Unexpectedly, we found that the averaged spike waveform hardly varied between these two subtypes of striatal units in either PD or dystonic patients (*Figure 3—figure supplement 2B* and *Figure 3—figure supplement 3B*, respectively). Besides, we did not find significant differences in the firing rate of the putative SPNs and TANs in PD (*Figure 3—figure supplement 2C and D*, first column) or dystonic (*Figure 3—figure supplement 3C and D*, first column) patients. However, the CV of the ISI was significantly smaller for the putative

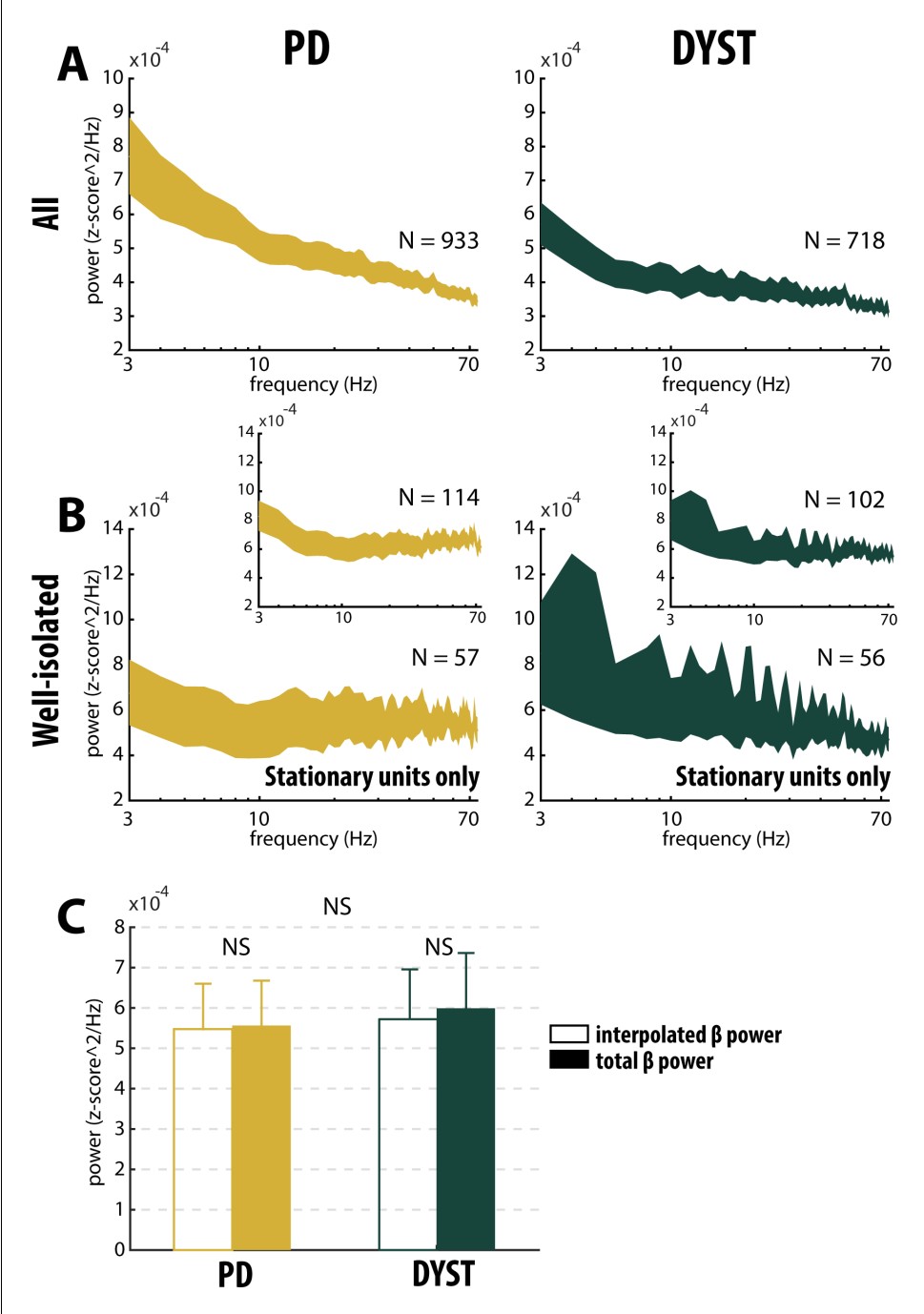

**Figure 7.** Absence of oscillatory spiking activity in the striatum of parkinsonian (PD) and dystonic (DYST) patients. Average (mean ± SEM) power spectrum densities (PSDs) of (**A**) all spiking activities and (**B**) only spiking activities recorded in the vicinity of well-isolated stationary units. Insets: Average PSDs of the spiking activities recorded in the vicinity of the well-isolated units (non-stationary and stationary units pooled). Since each band-pass filtered spiking signal was Z-score normalized (using its mean and SD) prior to PSD calculations, PSDs are expressed in z-score2/Hz. Abscissas are in log scale. N is the number of spiking activities averaged. (**C**) Average β (13–30 Hz) power of the spiking activities recorded in the vicinity of well-isolated stationary units. To interpolate the β power, PSDs were linearly interpolated [based on the two closest points that flanked the 13–30 Hz band - namely, the values at 12 and 31 Hz (spectral resolution of 1 Hz)]. Interpolated β power is the mean of the linearly interpolated values between 13 and 30 Hz. Total β power is the mean of the observed values between 13 and 30 Hz. Error bars represent SEMs. NS: non-significant (Wilcoxon signed rank test and Mann-Whitney U-test).

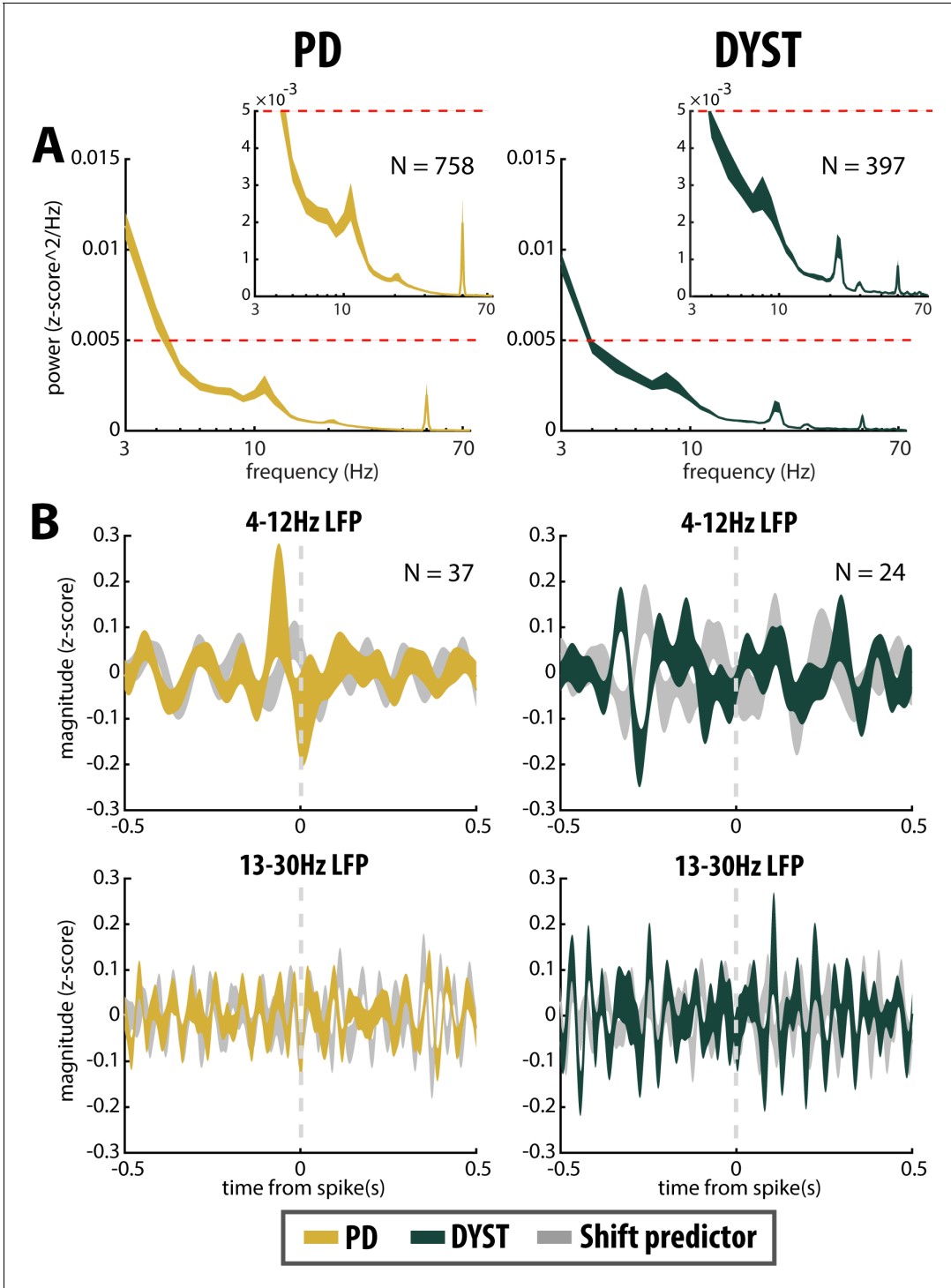

**Figure 8.** No locking between spike and β LFP oscillations in the striatum of parkinsonian (PD) and dystonic (DYST) patients. (**A**) Average (mean ± SEM) PSDs of striatal LFPs. In the insets, the ordinates are truncated for better visualization of the power. Abscissas are in log scale. N is the number of LFPs averaged. (**B**) Population (mean ± SEM) spike-triggered averages (STAs) of LFP. LFP was recorded in the vicinity of well-isolated stationary units (i.e., spiking activity and LFP were recorded on the same electrode) and offline band-pass filtered from 4 to 12 Hz (upper panels) or from 13 to 30 Hz (lower panels). For comparison, STAs-LFP were also calculated after randomly shifting the timestamp of each spike of the spike train [i.e., random time (comprised between 0 and 1 s) was added to the timestamp of each spike of the spike train] in order to abolish any relationship between LFP and spiking activity (Shift predictor). Dashed grey vertical lines indicate the time of the spikes (time = 0). N is the number of STAs-LFP averaged. Since each LFP was Z-score normalized (using its mean and SD) prior to PSD and STA calculations, PSDs and magnitudes of the STA-LFPs are expressed in z-score2/Hz and z-score, respectively.

TANs than for the putative SPNs in both diseases (*Figure 3—figure supplement 2C and D* and *Figure 3—figure supplement 3C and D*, second column), thus corroborating the view that TANs fire in a more regular fashion than the SPNs which are thought to be phasically activated (*Crutcher and DeLong, 1984*; *Deffains et al., 2010*; *Kimura et al., 1990*). The lack of significant differences in the spike waveform and the firing rate between our putative human SPNs and TANs can be accounted in several ways: misclassification of certain neurons, presence of other striatal interneurons (i.e., FSI) and/or differences in extracellular recording techniques (microelectrode type, position/placement, and impedance). Overall, it was difficult to distinguish the TANs from the SPNs in our human databases. Moreover, our spectral analyses did not reveal any significant differences between these two subtypes of striatal neurons in either disease (*Figure 3—figure supplement 2C and D*, and *Figure 3—figure supplement 3C and D*, third and fourth columns). Therefore, we considered our sorted striatal spiking activity to reflect SPN spiking activity and neglected the small fraction of TANs (which was probably smaller than our estimation).

## Outliers and cluster analysis of the striatal spiking features

Since we did not have a control condition or baseline levels, we could not identify the significant changes in the spiking features of the striatal neuronal activity. Nevertheless, we searched for outliers (i.e., data points that differ significantly from other observations) of each distribution. We examined the extreme values in the distributions and characterized the data points that could belong to a different population than the rest of the sample set. For each distribution, the values of the spiking feature (i.e., firing rate, CV of ISI, β power of the MUA or β power of STA-LFP)$\geq 2$ or $\leq -2$SDs of its mean value were considered outliers (empirical 68-95-99.7 rule). Remarkably, almost all of the outliers corresponded to feature values $\geq 2$ SDs of the mean value, thus indicating (as already observed) that the distributions of the values of the striatal spiking features were skewed to the right in both diseases. Nevertheless, we found that the proportions of outliers (regardless of the spiking feature and the isolation level) were small (*Figure 3—figure supplement 4*). Moreover, we did not find any significant disease effect on the proportions (chi-square test, p>0.05) or the values (Mann-Whitney U-test, p>0.05) of these outliers when only considering the well-isolated or well-isolated stationary units (*Figure 3—figure supplement 4*).

To further test whether two distinct SPN subpopulations could be observed in the current study, we performed 2D k-means cluster analysis with k = 2, using the discharge rate and the CV of the ISIs of each unit as input parameters (*Figure 9*).The basic idea behind this 2D k-means clustering was to define 2 clusters of data points so that the total within-cluster variation (i.e., the sum of the Squared Euclidean distances from data points to the cluster centroid) would be minimized within each of the two clusters. Using k = 2 as the predefined number of clusters systematically enforced the separation into two distinct SPN subpopulations. To assess how well-separated the two resulting clusters were, we calculated the silhouette values to measure how close each data point in one cluster was to data points in the other clusters. Silhouette values range from +1 (data points are very distant from other clusters), through 0 (data points are not distinctly in one cluster or another), to −1 (data points are probably assigned to the wrong cluster). The results showed that the two predefined clusters were not well-separated, regardless of the quality of the isolation or the stationarity of the striatal units (*Figure 9*, insets). In fact, in most of the scenarios, the second cluster contained a few data points with negative silhouette values, and if not, there were not enough data points to consider it a real cluster that represented a distinct and balanced SPN subpopulation. Equally important, in contrast with the view that discharge rate elevation and bursting activity coexist in the striatum of PD patients (*Singh et al., 2016*), the tendency to burst (CV of the ISIs > 1; *Kaneoke and Vitek, 1996*) decreased for units with a high discharge rate (*Figure 9*). Similar 2D or higher-order (3 or 4D) k-means cluster analyses (using the discharge rate, CV of the ISIs, MUA β power or STA-LFP β power as input parameters) also failed to reveal two well-separated clusters.

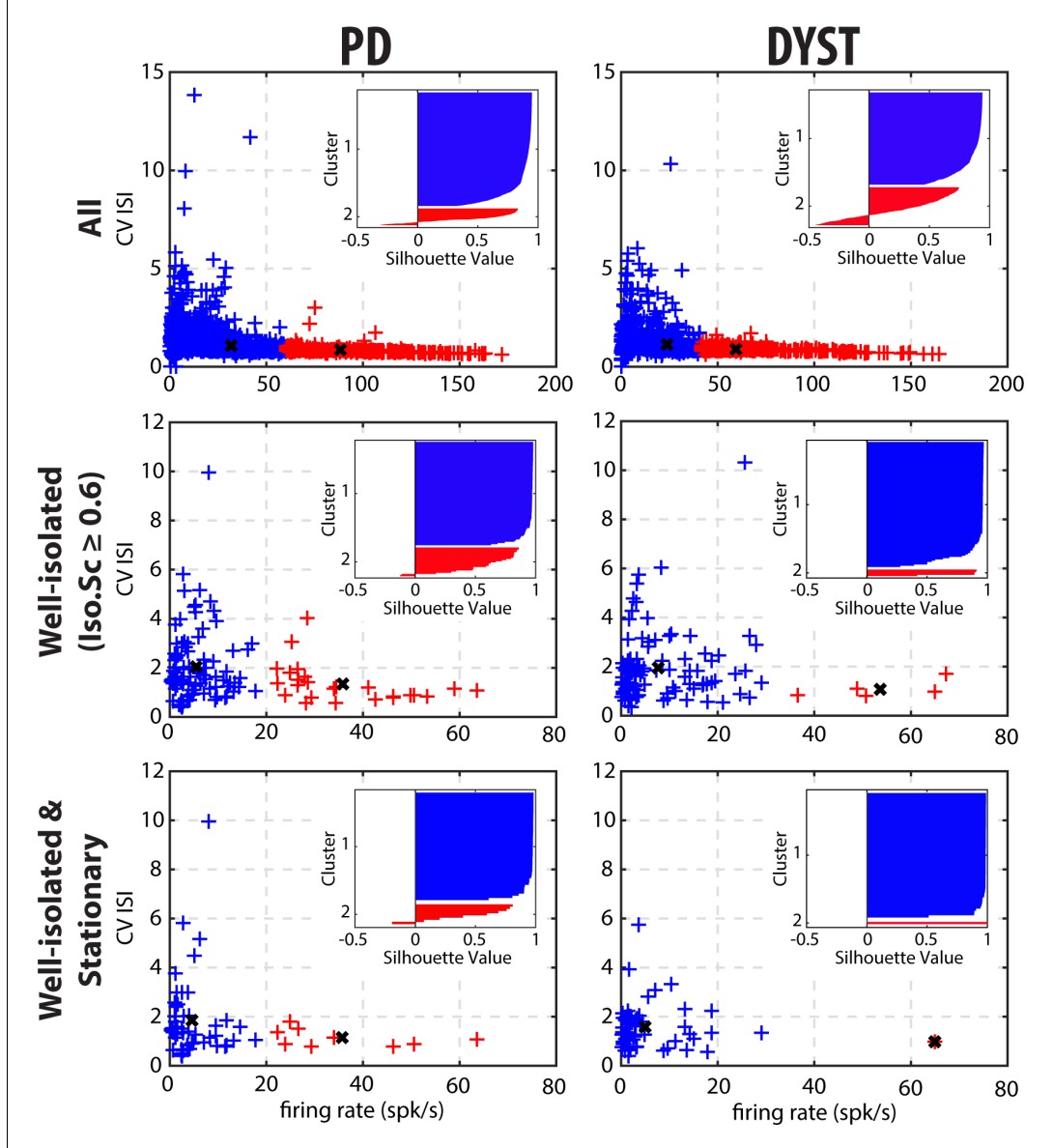

**Figure 9.** Cluster analysis using discharge properties does not reveal well-separated subpopulations of striatal units in parkinsonian (PD) and dystonic (DYST) patients. 2D k-means cluster analysis with k = 2, using the firing rate and the CV of the ISIs of each unit as input parameters. Analysis was performed when considering all (upper panels), only the well-isolated (middle panels) and only the well-isolated stationary (lower panels) units. Markers (x) represent cluster centroids. Inset: silhouette values were calculated for each clustering to assess how well-separated the two resulting clusters were. Silhouette values ranges from +1 (data points are very distant from other clusters), through 0 (data points are not distinctly in one cluster or another), to −1 (data points are assigned to the wrong cluster).

The online version of this article includes the following source data for figure 9:

**Source data 1.** 2D k-means cluster analysis for all units in PD.
**Source data 2.** 2D k-means cluster analysis for the well-isolated units in PD.
**Source data 3.** 2D k-means cluster analysis for the well-isolated and stationary units in PD.
**Source data 4.** 2D k-means cluster analysis for all units in Dystonia.
**Source data 5.** 2D k-means cluster analysis for the well-isolated units in Dystonia.
**Source data 6.** 2D k-means cluster analysis for the well-isolated and stationary units in Dystonia.

## Discussion

There is no consensus as to the impact of aberrant striatal signaling on the spontaneous discharge rate and pattern of the SPNs in PD and Dystonia (*Deffains et al., 2016*; *Ketzef et al., 2017*; *Liang et al., 2008*; *Mallet et al., 2006*; *Sharott et al., 2017*; *Singh et al., 2016*; *Singh et al., 2015*). Recordings of the activity of the SPNs are inherently difficult to perform given their very low firing rate and small size, leading to frequent loss and damage of these units (*DeLong, 1971*). Thus, a particular feature of striatal in vivo extracellular recordings is frequent 'injury activity'. These difficulties can distort the reported population averages of SPNs. For these reasons, we used an automated data-driven approach. A total of 933 and 718 MER segments from the posterior putamen [i.e., sensorimotor area of the striatum (*Parent and Hazrati, 1995*; *Yin and Knowlton, 2006*)] of PD and dystonic patients were analyzed. Here, striatal activity was not recorded in the caudate nucleus, but previous studies have shown that putaminal neurons fire at higher rate than the neurons of the caudate nucleus in PD patients (*Singh et al., 2016*). The striato-pallidal border was automatically detected using a machine learning algorithm (*Valsky et al., 2020*). MER segments were regularly spaced within the striatum and no manual search for units with low or high discharge rates was performed. Spike detection was performed offline with fully automatic quantification of the isolation quality and stationarity of the identified units. In contrast to previous reports (*Singh et al., 2016*), we found no drastic modulation in the SPN discharge rate and pattern (compared to values reported for normal/control animals or ET patients) in PD and dystonic patients.

### Low discharge rate and similar levels of striatal activity in PD and Dystonia

We found that the firing rate of the SPNs of both the PD and dystonic patients was extremely sensitive to the quality of the isolation of the units. Specifically, we found a negative correlation between striatal firing rate and isolation score (*Figure 2A and B*). The negative correlation between the discharge rate of the detected units and their isolation quality revealed that - unlike in the STN (*Deffains et al., 2014*) - the spike detection and sorting algorithm used in this study and/or the physical properties of the striatal neurons tended to erroneously classify noise events as spikes (false positives) rather than missing real spikes (false negatives). The amplitude of the extracellular action potentials (or the signal-to-noise ratio) is mainly sensitive to the impedance of the microelectrode and the distance between the recorded neurons and the microelectrode, and is not related to firing rate of the neurons. Therefore, we can rule out the possibility that fast firing striatal neurons were less frequently well-isolated and under-represented in the high-score bins. Removing the non-stationary units also reduced the striatal firing rate of SPNs in both the PD and dystonic patients (*Figure 3* and *Figure 3—figure supplement 1*, first column first column). This may indicate that the inclusion of non-stationary ('injured') units can also erroneously increase the striatal firing rate in PD and dystonic patients. Our automatic approach to spike detection, sorting and quality assessment revealed that the median firing rate of the striatal well-isolated stationary units was 3.85 and 2.55 Hz in PD and dystonic patients, respectively (*Figure 3* and *Figure 3—figure supplement 1*, first column).

Thus, our results are inconsistent with a recent study by *Singh et al., 2016* where the SPN discharge rate in PD ($30.2 \pm 1.2$ Hz) and dystonic ($9.3 \pm 0.6$ Hz) patients increased by ~15 and 5-fold compared to the low striatal discharge rate ($2.1 \pm 0.1$ Hz) found in patients suffering from ET (i.e., a disorder without any known BG malfunction). In the current study, we did not have a control condition, nor did we record the striatal activity from ET patients. Therefore, we cannot rule out the possibility of an increase in the striatal firing rate of both PD and dystonic patients compared to human controls. Moreover, the small fraction of striatal interneurons (including the TANs, presumably the cholinergic interneurons) that was probably included in our sample may have slightly distorted our results (*Figure 3* and *Figure 3—figure supplement 1*). Nevertheless, the putative TANs were hardly distinguishable from the putative SPNs in our human databases (*Figure 3—figure supplement 2* and *Figure 3—figure supplement 3*) and most of the striatal neurons are SPNs (*Graveland et al., 1985*; *Petryszyn et al., 2018*; *Petryszyn et al., 2014*). Finally, in the normal NHP, the discharge rate of the TANs is higher than that of the SPNs, so that their possible inclusion would shift our results towards a higher striatal discharge rate. In any case, we did not find any significant difference between PD and dystonic patients that might reflect a distinct level of striatal hyperactivity in PD

(*Figure 3* and *Figure 3—figure supplement 1*, first column). The different results between our study and *Singh et al., 2016* may be due to: (i) a difference between machine learning detection vs. manual electrophysiologist detection of the striato-pallidal border, (ii) different spike sorting and detection methods, (iii) the use of algorithms for the quantification of the isolation quality and stationarity of the units vs. manual electrophysiologist decision-making, and (iv) different recording techniques (microelectrode type, step size and regular spaced recordings vs. manual search of the recording sites).

## Lack of bursty and oscillatory pattern in the SPN spiking activity of PD and dystonic patients

The emergence of bursty and periodic oscillatory patterns after striatal dopamine depletion has been observed in other BG nuclei than the striatum (*Deffains et al., 2018*; *Deffains et al., 2016*; *Filion and Tremblay, 1991*; *Heimer et al., 2002*; *Raz et al., 2000*). Previous studies by our research group reported the emergence of oscillatory spiking activity in the TANs recorded in the striatum of MPTP-treated monkeys, thus indicating that abnormal oscillatory activity following striatal dopamine depletion did not spare the striatum (*Deffains et al., 2016*; *Raz et al., 1996*). However, we found no significant parkinsonism-related oscillations in SPN spiking activity (*Deffains et al., 2018*; *Deffains et al., 2016*). Here, in line with this finding, we did not observe the emergence of bursting (*Figure 3*, *Figure 3—figure supplement 1*, second column and *Figure 6*) or oscillatory (*Figure 3*, *Figure 3—figure supplement 1*, third column and *Figure 7*) spiking activity of the striatal units sorted in the PD and dystonic patients.

In contrast to our results, Singh and collaborators reported large increases in the bursting activity of the SPNs in PD and dystonic patients (*Singh et al., 2016*) and MPTP-treated monkeys (*Singh et al., 2015*). Levodopa-induced increases or decreases of the SPN firing rate may be the result of D1 or D2 receptor activation, respectively (*Hernandez-Lopez et al., 2000*; *Kitai and Surmeier, 1993*; *West and Grace, 2002*). Using the recognized responses to dopaminergic stimulation, *Singh et al., 2015* showed that most bursty SPNs in the parkinsonian NHP exhibited a D1 receptor response to levodopa, thus suggesting that the emergence of an abnormal firing pattern predominantly affected the D1 SPNs (i.e., the SPNs of the direct pathway). However, when using a single-cell juxtacellular recording-labeling technique in the 6OHDA rodent, *Sharott et al., 2017* showed that in addition to their excessive firing rate, D2 SPNs (that primarily compose the indirect pathway) displayed aberrant phase-locked burst firing to cortical β oscillations. In that study, it might be assumed that D2 SPNs, rather than D1 SPNs, would exhibit an abnormal firing rate and pattern in the dopamine-depleted striatum. Therefore, technical advances and further studies in PD patients and animal models should be carried out to reach a consensus.

As reported previously (*Deffains et al., 2018*; *Deffains et al., 2016*; *Kondabolu et al., 2016*; *Lemaire et al., 2012*; *Piña-Fuentes et al., 2018*; *Silberstein et al., 2003*; *Singh and Papa, 2019*), abnormal oscillations of the LFP have been recorded in the striatum of both PD and dystonic patients (*Figure 8A*). The LFP most likely represents the sub-threshold (e.g., synaptic input) activity, whereas the MUA reflects the efferent (output) activity of the local neuronal population (*Buzsáki et al., 2012*; *Logothetis, 2003*). In both diseases, striatal SPNs were therefore bombarded with pathological oscillations by their major afferent neurons, thus suggesting that the emergence of striatal β LFP oscillations is network-driven. Interestingly, other studies have shown that changes in the intrinsic properties of striatal neurons (i.e., increasing SPN excitability through amplification of striatal cholinergic tone) is sufficient to induce robust β LFP oscillations in the striatum (*McCarthy et al., 2011*). Due to the limitations of human extracellular recordings, we were not able to investigate the intrinsic properties of the striatal neurons and therefore cannot rule out the possibility that the intrinsic properties of the striatal neurons played a role in the emergence of β LFP oscillations in the striatum. In fact, it is likely that network-based and cell-autonomous mechanisms are not mutually exclusive. In any case, we found no significant phase-locking between striatal spiking activity and monopolar recorded LFP β oscillations (*Figure 3*, *Figure 3—figure supplement 1*, fourth column and *Figure 8B*), thus suggesting that the striatal oscillating synaptic inputs were probably not strong enough to entrain the spiking activity of the striatal SPNs. Indeed, the striatal SPNs failed to express pathological β oscillatory activity in either PD or Dystonia. Moreover, recent studies have demonstrated that monopolar and bipolar BG LFPs may be contaminated by the volume conductance of cortical electroencephalogram (EEG) activity (*Lalla et al., 2017*; *Marmor et al., 2017*).

Therefore, it is likely that BG LFPs, including the LFPs recorded in the striatum, do not accurately reflect local cellular activity and should be at best interpreted with caution. Exaggerated striatal LFP oscillations in PD and Dystonia cannot therefore be regarded as direct evidence for the presence of SPN spiking oscillatory activity.

### Is the imbalance in the activity of SPN subpopulations evident in PD and Dystonia?

Obviously, striatal activity must be affected in PD and dystonic patients. A popular theory is that the hypoactivity of the direct pathway (originating from D1 SPNs) and the hyperactivity of the indirect pathway (originating from D2 SPNs) lead to excessive GPi/SNr inhibitory inputs to the thalamus in PD (*Wichmann and DeLong, 2003*) and vice-versa in Dystonia (*Guehl et al., 2009*). These aberrant GPi/SNr inhibitory inputs to the thalamus lead to the release of abnormal output commands and result in the emergence of the clinical symptoms. Unlike in rodent studies [see e.g., (*Sharott et al., 2017*)], extracellular recordings of spontaneous activity cannot discriminate between the spiking activity of the striatal D1 and D2 SPNs. However, if striatal dopamine depletion drastically enhances the differences (in discharge rate and pattern) between the two distinct SPN subpopulations (but see *Ketzef et al., 2017*), one would expect to see a bimodal distribution of SPN discharge properties after dopamine depletion and/or distinct clusters of SPN activity in our patients. Instead, visual inspection of the distributions of the discharge rate (*Figure 5*) and the CVs of ISIs (*Figure 6A*) revealed long-tailed unimodal distributions in both diseases. Moreover, the cluster analysis using SPN discharge properties failed to identify two well-separated SPN subpopulations (*Figure 9*).

In conclusion, our results in patients extend our previous study in the MPTP NHP model of PD (*Deffains et al., 2016*) and studies in the 6-OHDA rodent model (e.g., *Ketzef et al., 2017*; *Maltese et al., 2019*) and demonstrate that abnormal activity along both the direct and indirect pathways of the BG network is not caused by drastic changes in spontaneous SPN spiking activity. Recently, *Maltese et al., 2019* reported abnormal recruitment (e.g., during behavior) of D1 and D2 SPNs (which are mostly silent at rest in the healthy condition) in parkinsonism that resulted in an aberrant net balance of striatal outputs, although there were no changes in the firing rate of the individual neurons. Our experimental approach (i.e., extracellular recordings in awake patients in the operating room) does not allow us to estimate the sample size of activated neurons. Moreover, we did not have a control condition and recordings were not performed during behavior to minimize surgical procedure time. Therefore, further experiments in primates (at least in monkeys, if not in patients) are needed to validate or refute this possibility. It is likely that these small to moderate changes in spontaneous SPN discharge are amplified by BG downstream structures (*Crompe et al., 2020*), thus leading to the clinical symptoms of PD and possibly of Dystonia.

## Materials and methods

### Patients and surgery

Patients with PD and Dystonia were recruited from the movement disorders clinics at the Hadassah Medical Center in Jerusalem. All patients were scheduled to undergo implantation of DBS electrodes into the GPi and underwent MR imaging, and evaluation for motor and non-motor impairments within 30 days prior to surgery. Data were collected from 16 PD and 13 dystonic (non-genetic and genetic Dystonia) patients. Patient demographic information appears in *Table 1*. Note that our DBS recordings in ET patients started 10 mm above the thalamic target and therefore did not include striatal recordings. All patients met the criteria for DBS and signed a written informed consent for surgery that involved microelectrode recordings. This study was authorized and approved by the Institutional Review Board of Hadassah Hospital in accordance with the Helsinki Declaration (reference code: 0168–10-HMO).

Surgery was performed using a CRW stereotactic frame (Radionics, Burlington, MA, USA). BG target coordinates were chosen as a composite of the indirect anterior commissure-posterior commissure (AC-PC) atlas-based location and direct (1.5 or 3Tesla) T2 magnetic resonance imaging, using Framelink 4 or 5 software (Medtronic, Minneapolis, USA). All recordings used in this study were made while the patients were fully awake (no sedation or anesthesia) and the PD patients were off dopaminergic medication (overnight washout >12 hr).

**Table 1.** Patient demographic information.

| Patient no. | Disease | Surgery side | Trajectories | Gender | Age at onset | Age at surgery | Disease duration (y) |
|---|---|---|---|---|---|---|---|
| 1 | PD | bilateral | R(2) ; L(1) | M | 40 | 62 | 22 |
| 2 | PD | unilateral | R(1) | F | 45 | 62 | 17 |
| 3 | PD | bilateral | R(2) ; L(2) | F | 50 | 62 | 12 |
| 4 | PD | unilateral | L(1) | F | 45 | 62 | 17 |
| 5 | PD | unilateral | R(2) | F | 49 | 60 | 11 |
| 6 | PD | bilateral | R(2) ; L(2) | F | 66 | 72 | 6 |
| 7 | PD | bilateral | R(2) ; L(1) | F | 43 | 59 | 16 |
| 8 | PD | bilateral | R(2) ; L(2) | F | 54 | 68 | 14 |
| 9 | PD | bilateral | R(2) ; L(2) | F | 55 | 62 | 7 |
| 10 | PD | bilateral | R(2) ; L(2) | F | 46 | 58 | 12 |
| 11 | PD | bilateral | R(2) ; L(2) | F | 41 | 57 | 16 |
| 12 | PD | bilateral | R(2) ; L(2) | M | 53 | 66 | 13 |
| 13 | PD | unilateral | L(2) | F | 53 | 63 | 10 |
| 14 | PD | bilateral | R(2) ; L(2) | M | 35 | 44 | 9 |
| 15 | PD | bilateral | R(1) ; L(1) | M | 41 | 57 | 16 |
| 16 | PD | unilateral | R(2) | M | 50 | 57 | 7 |
| 17 | Dystonia (NG) | bilateral | R(1) ; L(2) | F | 36 | 56 | 20 |
| 18 | Dystonia (NG) | bilateral | R(2) ; L(2) | M | 44 | 49 | 5 |
| 19 | Dystonia (NG) | bilateral | R(1) ; L(2) | F | 45 | 65 | 20 |
| 20 | Dystonia (NG) | bilateral | R(2) ; L(1) | F | 58 | 60 | 2 |
| 21 | Dystonia (NG) | bilateral | R(2) ; L(2) | F | 13 | 19 | 6 |
| 22 | Dystonia (NG) | bilateral | R(1) ; L(1) | F | 60 | 63 | 2 |
| 23 | Dystonia (NG) | bilateral | R(2) ; L(2) | M | 60 | 62 | 2 |
| 24 | Dystonia(NG) | bilateral | R(2) ; L(2) | F | 69 | 71 | 2 |
| 25 | Dystonia (G) | bilateral | R(2) ; L(2) | M | 16 | 25 | 9 |
| 26 | Dystonia (G) | bilateral | R(2) ; L(2) | M | 12 | 18 | 6 |
| 27 | Dystonia (G) | bilateral | R(1) ; L(2) | M | 24 | 39 | 15 |
| 28 | Dystonia (G) | bilateral | R(2) ; L(2) | M | 39 | 54 | 15 |
| 29 | Dystonia (G) | bilateral | R(1) ; L(2) | M | 53 | 56 | 3 |

PD patients (5 males and 11 females) were 60.7 ± 6.1 years old and with a disease duration of 12.8 ± 4.4 years (mean ± standard deviation, SD). Dystonic patients (7 males and six females) were 49.0 ± 18.0 years old and with a disease duration of 8.2 ± 6.9 years (mean ± standard deviation, SD). NG: non-genetic; G: genetic; R: right; L: left; Numbers in parentheses indicate the number of microelectrode trajectories; M: male; F: female.

## Data acquisition

The data were acquired using two systems: MicroGuide (prior to 2015, previously described [*Deffains et al., 2014*]) and Neuro Omega (from 2015, previously described [*Valsky et al., 2020*]).

## Microelectrode recordings

For every recording session, a microelectrode recording (MER) exploration using one or two micro-electrode trajectories (2 mm apart) was made starting 15 mm above the pre-operative T2 MRI image-based calculated target. Our trajectories followed a double-oblique approach (50 to 80° from the axial AC-PC plane and 0 to 10° degrees from the midsagittal plane) through the posterior puta-men and the GPe and towards the ventral border of the posterior-lateral portion of the GPi target (*Figure 1* and *Video 1*). The target coordinates were in the range of the approximate anatomic coordinates for the motor domain of the GPi: Lateral ($X$) = 19–22 mm from the midline or ~18 mm

from the third ventricle wall; Anterior (*Y*) = 1–3.5 mm from the mid-commissural point (MCP), and vertical (*Z*) = −1 to −4 below the AC-PC plane.

The 'central' electrode was directed at the ventral border of the posterior-lateral portion of the GPi and the 'anterior' electrode was located 2 mm anterior/ventral to the central electrode in the parasagittal plane. Some of these recordings were made by a single microelectrode trajectory (instead of two) to accommodate cortical anatomy under the burr hole and brain blood vessels (*Machado et al., 2006*).

MER segments were regularly sampled in space in order to avoid recording bias towards particular striatal units. For all our GPi-DBS surgeries, the step size between two MER segments ranged from 100 to 200 µm and was controlled by the neurophysiologist to achieve optimal identification of the pallidal borders. At each step, MER segments were recorded from 4 to 140 s (after a 2 s signal stabilization period).

A total of 93 microelectrode trajectories aiming at the GPi were analyzed (48 in PD and 45 in Dystonia), yielding a total of 933 and 718 MER segments within the posterior putamen of patients suffering from PD and Dystonia, respectively. Striatum-GPe borders were automatically detected using a machine learning software (*Valsky et al., 2020*).

## Group-based microelectrode track trajectories

The trajectories of GPi DBS patients were reconstructed using the open source LeadDBS program (www.lead-dbs.org) using LeadDBS v2.2.3 (*Horn et al., 2019*; *Horn and Kühn, 2015*). *Lead Group*, implemented within the Lead-DBS environment (*Treu et al., 2020*), was used to graphically illustrate the group-based microelectrode track trajectories within the posterior putamen. Pre-operative imaging was performed on a 1.5T or 3T MRI and included a pre-operative T1 and T2 sequence. All patients received post-operative CT. The post-operative CT was co-registered to anchor modality with the pre-operative T1 MRI, using a two-stage linear registration (rigid followed by affine) as implemented in Advanced Normalization Tools (*Avants et al., 2008*). Linear co-registration between the MR T1 and T2 modalities was done using SPM12 (*Penny et al., 2007*). Pre- (and post-) operative acquisitions were spatially normalized into MNI_ICBM_2009b_NLIN_ASYM space (*Fonov et al., 2011*) based on pre-operative acquisitions using the Symmetric Normalization (SyN) registration approach as implemented in Advanced Normalization Tools (*Avants et al., 2008*). Nonlinear deformation into template space was achieved in five stages. After two linear (rigid followed by affine) steps, a nonlinear (whole brain) SyN-registration stage was followed by two nonlinear SyN-registrations that consecutively focused on the GP as defined by subcortical masks in *Schönecker et al., 2009*. DBS electrode localizations were corrected for brain shift in post-operative acquisitions by applying a refined affine transform calculated between the pre- and post-operative acquisitions that were restricted to the GP as implemented in the brain shift-correction module of Lead-DBS software (*Horn and Kühn, 2015*). DBS electrodes were automatically pre-localized in native and template space using the PaCER algorithm (*Husch et al., 2018*). DBS electrodes were manually localized based on post-operative acquisitions using a tool specifically designed for this task.

A total of 36 microelectrode track trajectories (13 in PD and 23 in Dystonia) of four PD and six dystonic patients were visualized (*Figure 1*). We failed to include some patients because of corrupted or missing pre- or post-operative images. A 100 µm T1 scan of an ex-vivo human brain, acquired on a 7T MRI scanner (*Edlow et al., 2019*) served as a background template in *Figure 1*. The definition of the striatum, GPe and GPi boundaries was informed by the Distal atlas (*Ewert et al., 2018*). For further visualization of microelectrode track trajectories in the posterior putamen see *Video 1*.

## Offline spike sorting and assessment of isolation quality and stationarity of sorted units

To avoid human biases that could corrupt spike detection and sorting outcomes (*Pedreira et al., 2012*; *Rey et al., 2015*), single-unit striatal activity was assessed by sorting the spike trains from each MER segment recorded within the striatum using an automatic offline spike detection and sorting method (Offline Sorter v4.4.2.0, Plexon Inc, Dallas, Texas, RRID:SCR_000012). For each bandpass filtered spiking signal (multi-unit activity, MUA), spikes were detected using a negative voltage threshold trigger systematically set at 3SDs from the mean of its peak height histogram. Salient

features of the detected spikes were extracted with principal component analysis, using the scores of the first two principal components as features for clustering. Then, identification of different spike clusters reflecting the activity of different units was done by an automatic k-means clustering algorithm (K-Means Scan with a number of clusters between 1 and 5 and a circular seed cluster pattern for choosing the initial cluster centers). The clustering that produced the best value of the Pseudo-F statistic was then chosen as the clustering pattern for that band-pass filtered spiking signal. Finally, the isolation quality of the units identified by the spike sorter software was systematically graded by measuring their isolation scores using an independent algorithm (*Joshua et al., 2007*). The isolation score ranges from 0 (i.e. highly noisy) to 1 (i.e. perfect isolation).

To assess the stationarity of the firing rate and spike amplitude of each well-isolated unit (isolation score $\geq$0.6), we segmented the data comprised between the first and the last spike of its spike train into 10 equal non-overlapping time bins. Then, we Z-score normalized the firing rate and the average spike amplitude in each bin using the mean and the SD of either the firing rate or the average spike amplitude calculated over the 10 bins. Finally, the slope value of the linear regression line for the firing rate and the average spike amplitude were calculated and expressed in z-score/bin. Units with slope values of the linear regression line for the firing rate or average spike amplitude that were greater than or equal to the 70th percentile were defined as non-stationary units.

## Discharge pattern assessment of spike train

For each spike train of the sorted units, we calculated the inter-spike intervals (ISI) and generated their ISI histograms. In parallel, we also computed the autocorrelograms of the spike train of the well-isolated stationary units, calculated for ±500 ms offset with 10ms-bins. For each autocorrelogram, values were normalized so that autocorrelation values ranged from 0 to 1.

For burst detection, we applied the Poisson surprise method with the surprise maximization (SM) search algorithm (*Legéndy and Salcman, 1985*) to each spike train of the well-isolated stationary units, using the following parameters: minimal burst length = 3 spikes; threshold surprise value (S) = 10; burst ISI limit = mean (ISI)/2; add limit = 150% of burst ISI limit and inclusion criteria (IC) = 5.

For each spike train of the well-isolated stationary units, the frequency (number of bursts/s) and mean duration (s) of the burst episodes were calculated over their entire recording span. These two metrics were used to determine the burst prevalence for each unit which was defined as the burst frequency * mean burst duration. For each unit, the burst prevalence (range: 0–1) represents the probability that the discharge pattern is bursty.

## Power spectral density

Prior to power spectral density (PSD) calculations, each band-pass filtered spiking signal (multi-unit activity, MUA) recorded in the vicinity of a sorted unit was Z-score normalized (using its mean and SD) to obtain an unbiased estimate (by the electrode impedance, the A/D characteristics of the recording system, or the amplitude of the recorded neuronal activity) of the oscillatory activity (*Zaidel et al., 2010*). The Z-normalized signal was rectified by the 'absolute' operator (*Deffains et al., 2018*; *Deffains et al., 2016*; *Deffains et al., 2014*; *Moran et al., 2008*; *Zaidel et al., 2010*). The rectified signal follows the envelope of the MUA and therefore enables the detection of periodic oscillatory activities with frequencies below the range of the online band-pass filter. Since the LFP frequency domain was filtered out, the resulting PSD only represented the oscillatory features of the spiking activity. Although up to five units could be sorted from each MUA (see above in *Offline spike sorting and assessment of isolation quality and stationarity of sorted units*), each MUA was only analyzed once.

The PSD of each rectified Z-normalized signal (expressed in z-score2/Hz) was calculated using Welch's method with a 1 s Hamming window (50% overlap) and a spectral resolution of 1 Hz (nfft = 44000 or 48000, sampling frequency = 44 or 48 kHz depending on the acquisition system). To evaluate the β power, the baseline values in the 13–30 Hz range of each PSD were linearly interpolated [based on the two closest points that flanked the 13–30 Hz band, namely the values at 12 and 31 Hz (spectral resolution of 1 Hz, see above)] and averaged. Then, the β power (i.e., the mean of the observed values between 13 and 30 Hz) was compared to the interpolated β power (baseline).

Similarly, we also calculated the PSD of the striatal monopolar (0.1–300 Hz) LFP. To do so, the PSD of each Z-normalized LFP was calculated using Welch's method recorded in the vicinity of the cells (see above for the parameters; but nfft = 1375, sampling frequency = 1.375 kHz) and without prior rectification by the absolute operator.

## Spike-triggered average of the LFP

To investigate the spike-LFP relationship in the temporal domain, we also calculated the spike-triggered average (STA) of the LFPs (*Deffains et al., 2016*; *Goldberg et al., 2004*). To do so, the LFP was recorded in the vicinity of the sorted units (i.e., spiking activity and LFP were recorded on the same electrode). Each Z-normalized LFP was offline band-pass filtered from 4 to 12 Hz or from 13 to 30 Hz (4-pole Butterworth filter, filtfilt Matlab function). For comparison, STAs-LFP were also calculated after randomly shifting the timestamp of each spike of the spike train [i.e., random time (comprised between 0 and 1 s) was added to the timestamp of each spike of the spike train] in order to abolish any relationship between LFP and spiking activity (Shift predictor). The PSD of the STA-LFP was calculated as for the LFP.

## Software and statistics

Anatomical analysis was done with the open source LeadDBS program (www.lead-dbs.org). All the physiological data and statistical analyses were carried out using MATLAB R2016a routines (Mathworks, Natick, MA, USA, RRID:SCR_001622). Mann-Whitney U-tests and Wilcoxon signed rank tests were used for statistical comparisons of two unpaired and paired sample means, respectively. Statistical comparisons of two sample proportions were performed using Chi-square tests. Analysis of the subset of $\geq$10 s-MER segments with striatal well-isolated stationary units yielded similar results (*Figure 3—figure supplement 5* and *Figure 6—figure supplement 1*) to those reported here, thus indicating that our results were not confounded by the shortest MER segments. The criterion for statistical significance was set at $p < 0.05$ for all statistical tests.

# Acknowledgements

We thank Andreas Horn and Simon Oxenford for their assistance in illustrating the group-based microelectrode track trajectories using Lead-DBS software, Atira Bick for pre- and post-operative imaging, Esther Singer for editing the manuscript and the patients for agreement to participate in the study and authorizing the use of their electrophysiological recordings. This study was supported by the European Research Council (ERC), Rosetrees, Israel Science Foundation (ISF) and Israel Authority for Innovation grants to HB, and the French National Research Agency (ANR) and the French National Center for Scientific Research (CNRS) to MD.

# Additional information

### Funding

| Funder | Author |
| --- | --- |
| European Research Council | Hagai Bergman |
| Rosetrees | Hagai Bergman |
| Israel Science Foundation | Hagai Bergman |
| Israel Authority for Innovation | Hagai Bergman |
| French National Research Agency | Marc Deffains |
| French National Center for Scientific Research | Marc Deffains |

The funders had no role in study design, data collection and interpretation, or the decision to submit the work for publication.

## Author contributions
Dan Valsky, Conceptualization, Software, Formal analysis, Investigation, Methodology, Writing - original draft, Writing - review and editing; Shai Heiman Grosberg, Software, Formal analysis, Writing - review and editing; Zvi Israel, Conceptualization, Resources, Supervision, Project administration, Writing - review and editing; Thomas Boraud, Conceptualization, Writing - review and editing; Hagai Bergman, Conceptualization, Resources, Supervision, Funding acquisition, Investigation, Methodology, Writing - original draft, Project administration, Writing - review and editing; Marc Deffains, Conceptualization, Software, Formal analysis, Supervision, Investigation, Methodology, Writing - original draft, Project administration, Writing - review and editing

## Author ORCIDs
Dan Valsky (iD) https://orcid.org/0000-0002-2385-424X
Hagai Bergman (iD) http://orcid.org/0000-0002-2402-6673
Marc Deffains (iD) https://orcid.org/0000-0003-0734-6541

## Ethics
Human subjects: All patients met the criteria for DBS and signed a written informed consent for surgery that involved microelectrode recording. This study was authorized and approved by the Institutional Review Board of Hadassah Hospital in accordance with the Helsinki Declaration (reference code: 0168-10-HMO).

## Decision letter and Author response
Decision letter https://doi.org/10.7554/eLife.57445.sa1
Author response https://doi.org/10.7554/eLife.57445.sa2

## Additional files

### Supplementary files
• Transparent reporting form

### Data availability
All data generated or analysed during this study are included in the manuscript and supporting files. Source data files have been provided for Figures 2, 3 and 9.

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
