## [Decision Letter]

**Acceptance summary:**

A prevailing theory of basal ganglia dysfunction in Parkinson's disease is that it originates through changes in the firing rates or patterns of D2-expressing spiny projection neurons in the striatum. Contrary to long-standing assumptions, the present study finds no evidence for spiny neuron dysfunction across a large data set collected from human PD patients, and that spike sorting errors were the likely cause of perceived changes in neuronal firing rates in previous studies. These findings resolve a long-standing controversy about the nature of striatal dysfunction in PD and suggest that changes in basal ganglia pathophysiology likely arise through structures outside the striatum.

**Decision letter after peer review:**

Thank you for submitting your article "What is the true discharge rate and pattern of the striatal projection neurons in Parkinson's disease and Dystonia?" for consideration by *eLife*. Your article has been reviewed by three peer reviewers, including Aryn H Gittis as the Reviewing Editor and Reviewer #1, and the evaluation has been overseen by Richard Ivry as the Senior Editor. The following individual involved in review of your submission has agreed to reveal their identity: Robert S Turner (Reviewer #2).

The reviewers have discussed the reviews with one another and the Reviewing Editor has drafted this decision to help you prepare a revised submission.

Summary:

In this manuscript, Valsky et al. seek to clarify an important outstanding question regarding the pathophysiology of Parkinson's disease (PD). There is disagreement in existing literature as to whether the spiking activity of striatal projection neurons (SPNs) is grossly abnormal (as reported by Papas and colleagues) or not (as reported by several groups including some of the current authors). The authors recorded from SPNs in PD and dystonia patients during DBS implantation surgeries. The study is methodologically rigorous, using recordings from regularly spaced striatal locations, and highly automated recording and analysis procedures, thus avoiding sampling biases and the subjectivity of some data analysis steps. The authors show convincingly that firing rates and firing patterns of striatal neurons in PD patients do not differ significantly from those in patients with dystonia. Some disease-dependent differences were initially observed in firing rate and burst firing, but those differences disappeared when analysis was restricted to well-isolated stationary recordings, setting the bar for the level of rigor for data analysis that should be emulated by other groups. The recordings also showed no evidence of oscillatory firing (e.g., at β frequencies) or spike-field locking. Cluster analysis failed to yield evidence for the presence of subpopulations of SPNs. This is a significant study that may help us to better understand the discrepant literature results regarding striatal firing abnormalities in parkinsonism. The authors conclude that SPN activity is not grossly abnormal in PD and dystonia, as some investigators have reported. The finding that SPN neurons in PD patients do not show increased firing rates or changes in overall firing pattern challenges the long-standing theory that indirect pathway hyperactivity is caused by increased firing rates of D2-expressing SPNs and suggests that dopamine loss in parkinsonism may alter striatal processing more during active behaviours than at rest.

Essential revisions:

1) Rebalance the number of main figures vs. supplementary figures: The rationale for having only three figures, but many supplementary figures, is an odd choice. Supplementary Figure 2A-D should be incorporated in the manuscript as figures, not supplements.

2) Revise the recording location figure (Figure 1—figure supplement 1) to provide clearer information about recording locations. The study is not concerned with DBS lead placement (as shown in the figure), but with microelectrode recordings. It would be more useful to show example microelectrode track trajectories in the figure, or to simply state in the text that the trajectories penetrated mostly the putamen, as they apparently did. It definitely is not necessary (and quite confusing) to show projections on 5 different brain atlases in the same figure. Is there a concern that the very different results of Singh et al. came from a different striatal territory? It would be useful to many readers if the authors explained the importance of recording location and any differences in recording locations between the current study and those in previous studies. The manuscript states that recordings were obtained from the sensorimotor part of putamen, but Figure 1—figure supplement 1 does not illustrate this point adequately. The small-scale 3-dimensional images do not provide adequate insight to the specific parts of putamen sampled from. The LeadDBS package provides a way to illustrate all recording trajectories, not just one as appears to be shown in Figure 1—figure supplement 1. Why multiple atlases are shown in this figure is not clear. Given the double oblique approach described, it is likely that recording trajectories passed through the most dorsal-medial part of putamen, which in NHPs is the termination site for projections from SMA, not primary motor cortex. This does not raise serious problems for the impact of the current study, but the impact of the current study would be enhanced by better documentation of all recording locations.

3) Population-level comparisons vs. single units: Many of the results are presented as population medians/means. To exclude admittedly unlikely possibilities, it would be helpful to also test for significant effects in single units and then compare the frequencies of significant effects between populations. For example, was β oscillatory firing or STA-LFP significant in any individual SPN?

4) Clarification of analysis procedures:

– The methods and analysis strategy for Figure 3 was hard to follow by aficionados. Could the approach used in that figure be better explained for a lay audience?

– "All recorded units were assigned to 10 bins… ": clarify the binning procedure in the text. The figure legend implies that the binning covered all units that were greater than a certain isolation score, and not those units that had a specific score (as implied in the text).

– Spike sorting: automated spike sorting is pointed to as an important factor that distinguishes the present work from that of other groups. Thus, it seems appropriate to provide a more detailed description of that automated procedure. It is possible to use Offline Sorter in non-automated ways that open up the potential for bias.

– Figure 2 and accompanying text: The authors should explain their method of z-scoring the power spectral density plots.

– Figure 2—figure supplement 2: The Z-normalization procedure for firing rates and amplitudes is not clear. Which mean and standard deviation are used for the z-scoring (normalization)?

5) Although SPNs do not show significant phase locking to β oscillations, the authors state that in monkeys, TANs do show significant phase locking following MPTP treatment. TANs can be distinguished from SPNs based on waveform analysis. Can the authors use waveforms to isolate TANs from the human data set, and show that these neurons exhibit significant phase locking? This would strengthen their conclusions about the lack of phase locking observed in SPNs.

---

## [Author Response]

Essential revisions:1) Rebalance the number of main figures vs. supplementary figures: The rationale for having only three figures, but many supplementary figures, is an odd choice. Supplementary Figure 2A-D should be incorporated in the manuscript as figures, not supplements.

To comply with reviewers’ request, we rebalanced the number of main and supplementary figures in the revised version of the manuscript. The original Figure 2—figure supplements 2, 3, 4, 5 and 6 have been re-named and are now main figures. In the revised manuscript, we have 9 main and 6 supplementary figures, 1 table and 1 video.

2) Revise the recording location figure (Figure 1—figure supplement 1) to provide clearer information about recording locations. The study is not concerned with DBS lead placement (as shown in the figure), but with microelectrode recordings. It would be more useful to show example microelectrode track trajectories in the figure, or to simply state in the text that the trajectories penetrated mostly the putamen, as they apparently did. It definitely is not necessary (and quite confusing) to show projections on 5 different brain atlases in the same figure. Is there a concern that the very different results of Singh et al. came from a different striatal territory? It would be useful to many readers if the authors explained the importance of recording location and any differences in recording locations between the current study and those in previous studies. The manuscript states that recordings were obtained from the sensorimotor part of putamen, but Figure 1—figure supplement 1 does not illustrate this point adequately. The small-scale 3-dimensional images do not provide adequate insight to the specific parts of putamen sampled from. The LeadDBS package provides a way to illustrate all recording trajectories, not just one as appears to be shown in Figure 1—figure supplement 1. Why multiple atlases are shown in this figure is not clear. Given the double oblique approach described, it is likely that recording trajectories passed through the most dorsal-medial part of putamen, which in NHPs is the termination site for projections from SMA, not primary motor cortex. This does not raise serious problems for the impact of the current study, but the impact of the current study would be enhanced by better documentation of all recording locations.

We thank the reviewers for their constructive question and agree that the information about recording locations should be described more clearly. We now provide this information in the revised version of the manuscript. In response to this comment, we:

– Added new Figure 1 to depict the group-based microelectrode track trajectories of both PD and dystonia patients.

– Added a video within the main body of the article.

– Added text in the Results section describing the recording locations and in the Discussion section which state that we did not record in the caudate nucleus, but that previous studies have shown that putaminal neurons fire at higher rate than the neurons of the caudate nucleus in PD patients.

– Added text in the Materials and methods section describing the group-based microelectrode track trajectory analysis.

3) Population-level comparisons vs. single units: Many of the results are presented as population medians/means. To exclude admittedly unlikely possibilities, it would be helpful to also test for significant effects in single units and then compare the frequencies of significant effects between populations. For example, was β oscillatory firing or STA-LFP significant in any individual SPN?

We thank the reviewers for this comment. Since we did not have a control/healthy condition or baseline levels, we cannot identify significant changes. Nevertheless, we decided to find the “outliers” of each distribution and then compare them (in terms of proportion and value). For each distribution, the values of the parameter (i.e., firing rate, CV of ISI, β power of the MUA or β power of STA-LFP) ≥ 2 or ≤ -2SDs of its mean value were considered outliers (empirical 68-95-99.7 rule). Remarkably, only one non well-isolated unit (Iso.sc< 0.6) exhibited a parameter value (i.e. CV of ISI) ≤ -2SDs of the mean value of the parameter. In the subsequent analyses, we therefore only considered the outliers that corresponded to the parameter values ≥ 2SDs of the mean value of the parameter. The results showed that the proportions of outliers (regardless of the parameter and the isolation level) were small. Moreover, we did not find any significant disease effect on the proportions (chi-square test, p > 0.05) or the values (Mann-Whitney U-test, p > 0.05) of these outliers when only considering the well-isolated or well-isolated stationary units. These observations are now reported in the main text of the revised version of the manuscript and in one new figure (Figure 3—figure supplement 4).

4) Clarification of analysis procedures:– The methods and analysis strategy for Figure 3 was hard to follow by aficionados. Could the approach used in that figure be better explained for a lay audience?

We made the necessary changes in the main text of the revised version of the manuscript to clarify and better explain the cluster analysis. See subsection “Outliers and cluster analysis of the striatal spiking features”:

“To further test whether two distinct SPN subpopulations could be observed in the current study, we performed 2D k-means cluster analysis with k=2, using the discharge rate and the CV of the ISIs of each unit as input parameters (Figure 9).[…] Similar 2D or higher-order (3 or 4D) k-means cluster analyses (using the discharge rate, CV of the ISIs, MUA β power or STA-LFP β power as input parameters) also failed to reveal two well-separated clusters.”

– "All recorded units were assigned to 10 bins… ": clarify the binning procedure in the text. The figure legend implies that the binning covered all units that were greater than a certain isolation score, and not those units that had a specific score (as implied in the text).

To avoid any ambiguity, we rephrased this sentence as follows: “The median and mean discharge rates of all the recorded units with an isolation score greater than or equal to one of the 10 evenly spaced values between 0 and 0.9 were calculated (Figure 2B).”

– Spike sorting: automated spike sorting is pointed to as an important factor that distinguishes the present work from that of other groups. Thus, it seems appropriate to provide a more detailed description of that automated procedure. It is possible to use Offline Sorter in non-automated ways that open up the potential for bias.

We thank the reviewers for raising this important issue. It is true that spike detection and sorting outcomes can vary from one experimenter to another if the procedure is not fully automatic. In fact, in most applications, the amplitude thresholds can be set manually for spike detection and the boundaries for the different clusters can also be manually delineated. Equally important, a posteriori visual inspection of the waveforms of the sorted units is commonly carried out to add or remove specific spikes (i.e. subjective “cleaning” of the spike train). Such approaches are problematic because they depend on subjective individual decision-making and may lead to potential errors (Pedreira et al., 2012; Rey et al., 2015); namely erroneously classifying noise events as spikes (false positives) or missing real spikes (false negatives). To avoid these human biases that could jeopardize our results, single-unit striatal activity was assessed by sorting spike trains from each MER segment recorded within the striatum using an automatic offline spike detection and sorting method (Offline Sorter v4.4.2.0, Plexon Inc, Dallas, Texas). For each band-pass filtered spiking signal (multi-unit activity, MUA), spikes were detected using a negative voltage threshold trigger systematically set at 3SDs from the mean of its peak height histogram. Salient features of the detected spikes were extracted with principal component analysis, using the scores of the first two principal components as features for clustering. Then, identification of different spike clusters reflecting the activity of different units was done by an automatic k-means clustering algorithm (K-Means Scan with a number of clusters between 1 and 5 and a circular seed cluster pattern for choosing the initial cluster centers). The clustering that produced the best value of the Pseudo-F statistic was then chosen as the clustering pattern for that band-pass filtered spiking signal. Finally, the isolation quality of the units identified by the spike sorter software was systematically graded by measuring their isolation scores using an independent algorithm (Joshua et al., 2007). The isolation score ranges from 0 (i.e. highly noisy) to 1 (i.e. perfect isolation). We added this more detailed description of our spike detection and sorting method to the Materials and methods section of the revised version of the manuscript.

– Figure 2 and accompanying text: The authors should explain their method of z-scoring the power spectral density plots.

We agree with the reviewers that we should clarify how the β power was expressed in z-score^2/Hz. Here, we did not Z-normalize the PSD plots themselves. In fact, each band-pass filtered spiking signal (MUA) recorded in the vicinity of a sorted unit was Z-score normalized prior to PSD calculations (Deffains et al., 2014). The Z-score normalization (or standardization) of each signal was performed using its mean and SD. This procedure is now described both in the Materials and methods and in the legend of Figure 3.

– Figure 2—figure supplement 2: The Z-normalization procedure for firing rates and amplitudes is not clear. Which mean and standard deviation are used for the z-scoring (normalization)?

We thank the reviewers for their constructive comment and agree that the Z-score normalization procedure for firing rates and amplitudes was not clear in the text of our original manuscript. For each spike train, we segmented the data comprised between the first and the last spike of the spike train into 10 equal non-overlapping time bins. Then, we calculated the firing rate and the average spike amplitude in each bin. Finally, to compare patients and diseases, we Z-score normalized the firing rate and the average spike amplitude in each bin, using the mean and the SD of either the firing rate or the average spike amplitude calculated over the 10 bins.

5) Although SPNs do not show significant phase locking to β oscillations, the authors state that in monkeys, TANs do show significant phase locking following MPTP treatment. TANs can be distinguished from SPNs based on waveform analysis. Can the authors use waveforms to isolate TANs from the human data set, and show that these neurons exhibit significant phase locking? This would strengthen their conclusions about the lack of phase locking observed in SPNs.

We agree with the reviewers that in our previous study in NHPs, we found that the spiking activity of the striatal TANs was synchronized (phase locked) to the LFP recorded in their surrounding areas. To address this issue in the current study, TANs were identified by 3 experts (DV, HB and MD) from the well-isolated and stationary striatal units included in both the PD and Dystonia databases. This identification was based on offline visual inspection of their electrophysiological features, including their spike waveforms. The 3D-classifications of the putative SPNs and TANs, using their firing rate, the CV of their ISI and the score of the first principal component (PC1) of their spike waveform did not clearly discriminate them in either PD or dystonic patients (Figure 3—figure supplement 2A and Figure 3—figure supplement 3A, respectively). The same 3D-classification, but using the score of PC2 or PC3 (instead of PC1) yielded the same qualitative results. Retrospectively, we found that, contrary to our expectations, the averaged spike waveform hardly varied between these two subtypes of striatal units in either the PD or dystonic patients (Figure 3—figure supplement 2B and Figure 3—figure supplement 3B, respectively). Moreover, we did not find significant differences in the firing rate of the putative SPNs and TANs in PD (Figure 3—figure supplement 2C and D, first column) or dystonic (Figure 3—figure supplement 3C and D, first column) patients. However, the CV of the ISI was significantly smaller for the putative TANs than for the putative SPNs in both diseases (Figures 3—figure supplement 2 and Figure 3—figure supplement 3C and D, second column), thus corroborating the view that TANs fire in a more regular fashion than the SPNs that are thought to be phasically activated (Crutcher and DeLong, 1984; Deffains et al., 2010; Kimura et al., 1990). Alternatively, experts might be more biased by the discharge pattern of the units. The lack of significant differences in the spike waveform and the firing rate between our putative human SPNs and TANs can be accounted for in several ways: misclassification of certain neurons, presence of other striatal interneurons (i.e., FSI) and different extracellular recording techniques (microelectrode type, position/placement, and impedance). Overall, it was difficult to distinguish the TANs from the SPNs in our human databases. Since our spectral analyses did not reveal any significant differences between these two subtypes of striatal neurons in either disease (Figures 3—figure supplement 2 and Figure 3—figure supplement 3C and D, third and fourth columns), we therefore decided not to exclude the fraction of putative TANs probably included in our samples.

Nevertheless, these spectral results should be interpreted with caution due to our inability to reliably identify the TANs in our human databases and in fact we cannot rule out the possibility that a small fraction of TANs probably included in our samples (which is most likely smaller than what we estimated) may have slightly distorted our results. We now address this important issue in the main text of the revised version of the manuscript and in the two new figures (Figure 3—figure supplement 2 and Figure 3—figure supplement 3).